# *SPLITZ*: Certifiable Robustness via Split Lipschitz Randomized Smoothing

## Abstract

Certifiable robustness gives the guarantee that small perturbations around an input to a classifier will not change the prediction. There are two approaches to provide certifiable robustness to adversarial examples– a) explicitly training classifiers with small Lipschitz constants, and b) Randomized smoothing, which adds random noise to the input to create a smooth classifier. We propose *SPLITZ*, a practical and novel approach which leverages the synergistic benefits of both the above ideas into a single framework. Our main idea is to *split* a classifier into two halves, constrain the Lipschitz constant of the first half, and smooth the second half via randomization. Motivation for *SPLITZ* comes from the observation that many standard deep networks exhibit heterogeneity in Lipschitz constants across layers. *SPLITZ* can exploit this heterogeneity while inheriting the scalability of randomized smoothing. We present a principled approach to train *SPLITZ* and provide theoretical analysis to derive certified robustness guarantees. We present a comprehensive comparison of robustness-accuracy tradeoffs and show that *SPLITZ* consistently improves upon existing state-of-the-art approaches on MNIST, CIFAR-10 and ImageNet datasets. For instance, with $\ell_2$ norm perturbation budget of $\epsilon = 1$, *SPLITZ* achieves **61.7%** top-1 test accuracy on CIFAR-10 dataset compared to state-of-art top-1 test accuracy 39.8%, a 55.0% improvement in certified accuracy over various approaches (including, denoising based methods, ensemble methods, and adversarial smoothing).

## 1 Introduction

As deep learning becomes dominant in many important areas, ensuring robustness during test time becomes increasingly important. Deep neural networks are vulnerable to small perturbations in the inputs leading neural networks to make wrong decisions (Huang et al., 2021; Salman et al., 2019; Jeong et al., 2021). Although many works have proposed heuristic defenses for training robust classifiers, they are often shown to be inadequate against adaptive attacks. Therefore, a growing literature on certifiable robustness has emerged; where the classifier's prediction *must be provably robust* around any input within a perturbation budget. There are two broad approaches to design classifiers which are certifiably robust: a) design classifiers which are inherently stable (i.e., smaller Lipschitz constants) (Gowal et al., 2018; Mirman et al., 2018; Lee et al., 2020). There are a variety of methods to train classifiers while keeping the Lipschitz constants bounded. The second approach is b) randomized smoothing (RS) (Cohen et al., 2019; Jeong et al., 2021; Lecuyer et al., 2019); here, the idea is to smooth the decision of a base classifier by adding noise at the input. The approach of RS has been generalized in several directions: Salman et al. (Salman et al., 2020) and Carlini et al. (Carlini et al., 2023) combine denoising mechanisms with smoothed classifiers, Salman et al. (Salman et al., 2019) combine adversarial training with smoothed classifiers, Zhai et al. (Zhai et al., 2020) propose a regularization which maximize the approximate certified radius and Horváth et al. (Horváth et al., 2021) combine ensemble models with smoothed classifiers.

**Pros and Cons of Lipschitz constrained training versus RS:** Lipschitz constrained training is often only feasible for smaller neural networks (with few layers) and provides deterministic guarantees on certified radius. The main challenge is that accurate estimation of Lipschitz constants becomes infeasible for larger networks, and upper bounds become loose leading to vacuous bounds on certified radius. RS on the other hand offers scalability to arbitrarily large networks and provide closed-form certified robust radius. These guarantees however, are probabilistic in nature and the smoothing procedure treats the entire classifier as a black box.

| Method | Extra data | Certified Test Accuracy at $\epsilon$ (%) | | | | |
|---|---|---|---|---|---|---|
| | | 1.50 | 1.75 | 2.00 | 2.25 | 2.50 |
| RS (Cohen et al., 2019) | ✗ | 67.3 | 46.2 | 32.5 | 19.7 | 10.9 |
| MACER (Zhai et al., 2020) | ✗ | 73.0 | 50.0 | 36.0 | 28.0 | - |
| Consistency (Jeong & Shin, 2020) | ✗ | 82.2 | 70.5 | 45.5 | 37.2 | 28.0 |
| SmoothMix (Jeong et al., 2021) | ✗ | 81.8 | 70.7 | 44.9 | 37.1 | 29.3 |
| DRT (Yang et al., 2021) | ✗ | 83.3 | 69.6 | 48.3 | 40.3 | 34.8 |
| *SPLITZ* (this paper) | ✗ | **94.5** | **93.0** | **91.7** | **90.1** | **88.2** |

Table 1: Comparison of certified test accuracy (%) on MNIST under $\ell_2$ norm perturbation. Each entry lists the certified accuracy using numbers taken from respective papers (RS results follow from previous benchmark papers (Jeong & Shin, 2020; Jeong et al., 2021)).

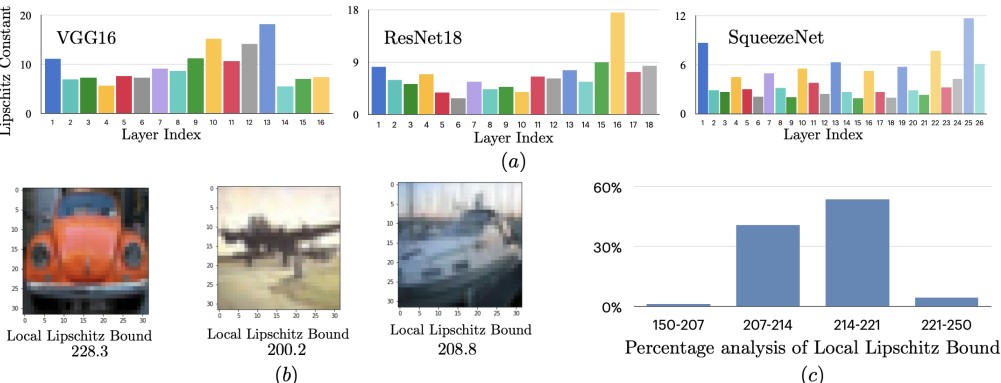

Figure 1: (a) Lipschitz constants of each affine layer in pretrained models: VGG16 (Simonyan & Zisserman, 2015), ResNet18 (He et al., 2016), SqueezeNet (Iandola et al., 2016). (b) Local Lipschitz (upper) bound for three random CIFAR-10 images on VGG16; (c) Percentage analysis of local Lipschitz (upper) bound in CIFAR-10 test data (additional results in Appendix A).

**Overview of *SPLITZ* and Contributions**. In this paper, we propose *SPLITZ*, which combines and leverages the synergies offered by both Lipschitz constrained training and randomized smoothing. The general idea is to split a classifier into two halves: the first half (usually a few layers) is constrained to keep a smaller Lipschitz (upper) bound, and the latter half of the network is *smoothed* via randomization. Interestingly, this approach yields state-of-the-art results for several datasets. As shown in Table 1, *SPLITZ* outperforms state-of-art techniques for every value of $\epsilon$ on the MNIST dataset. For $\epsilon$ ($\ell_2$ norm perturbation size) even as large as 2.5, where the state-of-the-art accuracy is 34.8%, *SPLITZ* achieves certified accuracy of around 88.2%. In Section 4, we present comprehensive set of results on MNIST, CIFAR-10 and ImageNet datasets. We also provide the comprehensive theoretical analysis of the certified robustness guarantee of *SPLITZ* classifier. [1]

**Intuition behind *SPLITZ***. The intuition behind *SPLITZ* comes from the following key observations: a) *Layer-wise Heterogeneity*: many modern deep networks exhibit heterogeneity in Lipschitz constants across layers. Fig 1(a) shows the per-layer Lipschitz constants for three networks (VGG16, ResNet18 and SqueezeNet). We observe that the values can vary widely across the layers, and quite often, latter half of the network often shows larger Lipschitz constants. b) *Input (local) heterogeneity*: We show the local Lipschitz (upper) bounds for three randomly sampled images from CIFAR-10 when passed through the first four layers of VGG16; note that the values of local Lipschitz bound can vary across different inputs (images). The same behavior across the entire CIFAR-10 test dataset is shown in Fig. 1(c). These observations motivate *SPLITZ* as follows: smoothing the input directly may not be the optimal approach as it does not account for this heterogeneity. Instead, by introducing noise at an intermediate stage of the classifier, the model can become more resilient to perturbations. This suggests the idea of splitting the classifier. Simultaneously, the first half of the network should also be "stable", which motivates constraining the Lipschitz bound of first half of the network.

---

[1] All of our codes are provided in the supplementary materials.

## 2 PRELIMINARIES ON CERTIFIED ROBUSTNESS

We consider a robust training problem for multi-class supervised classification, where we are given a dataset of size $N$, $\{x_i, y_i\}_{i=1}^N$, where $x_i \in \mathbb{R}^d$ denotes the set of features of the $i$th training sample, and $y_i \in \mathcal{Y} := \{1, 2, \ldots, C\}$ represents the corresponding true label. We use $f$ to denote a classifier, which is a mapping $f \colon \mathbb{R}^d \to \mathcal{Y}$ from input data space to output labels. From the scope of this paper, our goal is to learn a classifier which satisfies certified robustness, as defined next.

**Definition 1.** *(Certified Robustness) A (randomized) classifier $f$ satisfies $(\epsilon, \alpha)$ certified robustness if for any input $x$, we have*

$$\mathbb{P}(f(x) = f(x')) \geq 1 - \alpha, \; \forall x', \text{ such that } x' = x + \delta, \| \delta \|_p \leq \epsilon$$

*where the probability above is computed w.r.t. randomness of the classifier $f$.*
Intuitively, certified robustness requires that for any test input $x$, the classifier's decision remains locally invariant, i.e., for all $\forall x'$ around $x$, such that $\| x' - x \|_p \leq \epsilon$, $f(x) = f(x')$ with a high probability. Thus, $\epsilon$ is referred to as the certified radius, and $(1 - \alpha)$ measures the confidence. We mainly focus on $\ell_2$ norm ($p = 2$) for the scope of this paper.

The literature on certified robustness has largely evolved around two distinct techniques: *Randomized Smoothing* and *Lipschitz constrained training for Certifiably Robustness*. We first briefly summarize and give an overview of these two frameworks, before presenting our proposed approach of *Split Lipschitz Smoothing*.

**Randomized Smoothing (RS)** (Cohen et al., 2019) is a general procedure, which takes an arbitrary classifier (base classifier) $f$, and converts it into a "smooth" version classifier (smooth classifier). Most importantly, the smooth classifier preserves nice certified robustness property and provides easily computed closed-form certified radius. Specifically, a general smooth classifier $g_{RS}(\cdot)$ derived from $f$ is given as:

$$g_{RS}(x) = \underset{c \in \mathcal{Y}}{\operatorname{argmax}} \; \underset{\delta \sim \mathcal{N}(0, \sigma^2 I)}{\mathbb{P}} (f(x + \delta) = c) \tag{1}$$

Intuitively, for an input $x$, $g(x)$ will output the most probable class predicted by the base classifier $f$ in the neighbourhood of $x$ with a high confidence $1 - \alpha$. In the paper (Cohen et al., 2019), they prove that $g(x)$ is robust against $\ell_2$ perturbation ball of radius $\epsilon = \sigma \Phi^{-1}(p_A)$ around $x$, where $\sigma$ is the standard deviation of the Gaussian noise, and $p_A$ is the probability that the most probable class predicted by the classifier $f$ is $c_A$. RS is arguably the only certified defense which can scale to large image classification datasets. Based on RS, a number of studies have been undertaken in this field: RS was originally proposed to deal with $\ell_2$ norm bounded perturbations; but was subsequently extended to other norms using different smoothing distributions, including $\ell_0$ norm with a discrete distribution (Lee et al., 2019), $\ell_1$ norm with a Laplace distribution (Teng et al., 2020), and the $\ell_\infty$ norm with a generalized Gaussian distribution (Zhang et al., 2020). Other generalizations include combining RS with adversarial training to further improve certified robustness and generalization performance (Salman et al., 2019) or denoising mechanisms (such as diffusion models) are often considered in conjunction with RS (Salman et al., 2020; Carlini et al., 2023).

Achieving a *large* certified radius can be equivalently viewed as learning a classifier with *small* Lipschitz constant. The Lipschitz constant is a fundamental factor in numerous studies focused on training a certifiably robust neural network, which can be defined as follows:

**Definition 2.** *(Global and Local Lipschitz Constant(s)) For a function $f \colon \mathbb{R}^d \to \mathcal{Y}$, the Global, Local, and $\gamma$-Local Lipschitz constants (at an input $x$) are respectively, defined as follows:*

$$\text{(Global Lipschitz constant)} \; L_f = \sup_{x, y \in dom(f); x \neq y} \frac{\|f(y) - f(x)\|_p}{\|y - x\|_p} \tag{2}$$

$$\text{(Local Lipschitz constant)} \; L_f(x) = \sup_{y \in dom(f); y \neq x} \frac{\|f(y) - f(x)\|_p}{\|y - x\|_p} \tag{3}$$

$$\text{($\gamma$-Local Lipschitz constant)} \; L_f^{(\gamma)}(x) = \sup_{y \in B(x, \gamma); y \neq x} \frac{\|f(y) - f(x)\|_p}{\|y - x\|_p}, \tag{4}$$

*where $B(x, \gamma)$ denotes the $\ell_p$-ball around $x$ of radius $\gamma$, i.e., $B(x, \gamma) = \{u : \|u - x\|_p \leq \gamma\}$.*
Informally, $L_f^{(\gamma)}(x)$ captures the stability of the function $f$ in the neighborhood of $x$, where the neighborhood is characterized by an $\ell_p$-ball of radius $\gamma$.

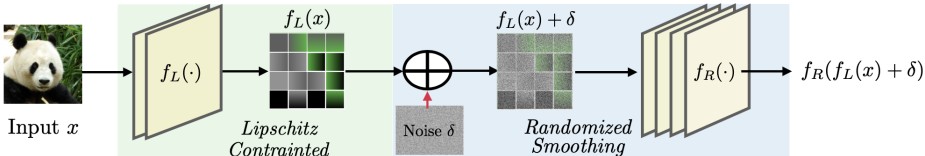

Figure 2: Schematic of *SPLITZ* training framework.

**Lipschitz constrained training for Certifiably Robustness** A reliable upper bound for the local Lipschitz constant is essential for the robustness of a classifier. However, computing the exact value of local Lipschitz constants can be computationally challenging, prompting researchers to seek approximations, in terms of upper bounds. Thus, a line of works focus on deriving a tighter local Lipschitz bound e.g.,Zhang et al. (2019); Fazlyab et al. (2019); Jordan & Dimakis (2020). Another line of works utilize the local Lipschitz bound to obtain better robustness guarantees, e.g., Hein & Andriushchenko (2017); Weng et al. (2018). Furthermore, there are several works which aim to train a certified robust classifier as we briefly summarize next. One approach is to estimate/upper bound the global Lipschitz constant of the classifier (during each training epoch) and use it to ensure robustness. For instance, Tsuzuku et al. (2018); Lee et al. (2020); Leino et al. (2021) follow this general approach. The challenge is that the bounds on global Lipschitz constants can be quite large, and do not necessarily translate to improve certified robustness. An alternative approach is to use a local Lipschitz bound (for each individual input $x$), as in Huang et al. (2021) and then explicitly minimize the Lipschitz bound during the training process. For simplicity, we refer to the upper bound of the local Lipschitz constant as the "local Lipschitz constant".

## 3  *SPLITZ*: INFERENCE, CERTIFICATION AND TRAINING

In this Section, we first describe the details of the proposed *SPLITZ* classifier along with the motivation as well as key distinctions from prior work. We then present new theoretical results on certified radius for *SPLITZ*. Subsequently, we describe the training methodology for *SPLITZ* as well as inference and computation of the certified radius. Suppose we are given a base classifier $f : \mathbb{R}^d \to \mathcal{Y}$ which is a composition of $K$ functions. Consider an arbitrary "split" of $f$ as $f(\cdot) = f_R(f_L(\cdot)) \triangleq f_R \circ f_L$. As an example, if the classifier has $K = 2$ hidden layers, i.e., $f(x) = f_2(f_1(x))$, then there are $K + 1 = 3$ possible compositions/splits: a) $f_R = I$, $f_L = f_2 \circ f_1$, b) $f_R = f_2$, $f_L = f_1$, and c) $f_R = f_2 \circ f_1$, $f_L = I$, where $I$ represents the identity function.

**Definition 3.** *(SPLITZ Classifier) Let $f$ be a base classifier: $\mathbb{R}^d \to \mathcal{Y}$. Consider an arbitrary split of $f$ as $f(\cdot) = f_R(f_L(\cdot))$. We define the SPLITZ classifier $g_{SPLITZ}(\cdot)$ as follows:*

$$g_{SPLITZ}(x) = \underset{c \in \mathcal{Y}}{argmax} \underset{\delta \sim \mathcal{N}(0,\sigma^2 I)}{\mathbb{P}} (f_R(f_L(x) + \delta) = c) \tag{5}$$

The *SPLITZ* smoothing classifier is illustrated in Fig 2. The basic idea of *SPLITZ* is two fold: smooth the *right half* of the network using randomized smoothing and constrain the Lipschitz constant of the *left half*. Specifically, to robustly classify an input $x$, we add noise to the output of the left half (equivalently, input to the right half) of the network, i.e., $f_L(x)$ and then follow the same strategy as randomized smoothing thereafter. While RS takes care of smoothing the right half, we would still like the left half to be as *stable* as possible. Thus in addition to smoothing, we need to ensure that the Lipschitz constant of the left half $f_L$ of the network is also kept small. We next present our main theoretical result, which allows us to compute the certified radius for *SPLITZ*.

**Theorem 1.** *Let us denote $L_{f_L}^{(\gamma)}(x)$ as the $\gamma$-local Lipschitz constant of the function $f_L$ at $x$ in a ball of size $\gamma$, and $R_{f_R}(f_L(x))$ as the certified radius of the function $f_R$ at the input $f_L(x)$, with probability at least $(1 - \alpha)$. Then, for any input $x$, with probability $1 - \alpha$, $g_{SPLITZ}(x)$ has a certified radius of at least,*

$$R_{g_{SPLITZ}}(x) = \max_{\gamma \geq 0} \ \min \left\{ \frac{R_{f_R}(f_L(x))}{L_{f_L}^{(\gamma)}(x)}, \gamma \right\} \tag{6}$$

The proof of Theorem 1 is presented in Appendix C. Given an input $x$, in order to compute the certified radius for SPLITZ classifier, we need $L_{f_L}^{(\gamma)}(x)$, i.e., the $\gamma$-local Lipschitz constant (discussed

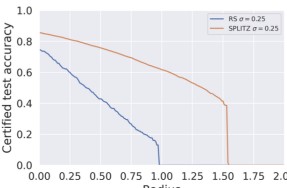 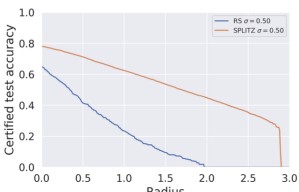 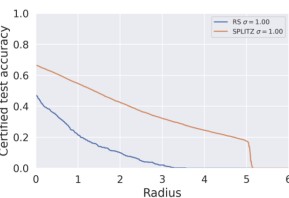

Figure 3: Comparison of certified radius with $\ell_2$ norm perturbation w.r.t *RS* (Cohen et al., 2019) and *SPLITZ* (ours), when varying levels of Gaussian noise $\sigma \in \{0.25, 0.5, 1.0\}$.

in the next Section) and the certified radius of right half of the classifier, i.e., $R_{f_R}(f_L(x))$. For Gaussian noise perturbation in the second half, $R_{f_R}(f_L(x))$ is exactly the randomized smoothing $\ell_2$ radius (Cohen et al., 2019), given as $R_{f_R}(f_L(x)) = \frac{\sigma}{2}(\Phi^{-1}(\underline{p_A}) - \Phi^{-1}(\overline{p_B}))$, where $\underline{p_A}$ is the lower bound of the probability that the most probable class $c_A$ is returned, $\underline{p_B}$ is upper bound of the probability that the "runner-up" class $c_B$ is returned.

**Remark 1: Optimization over $\gamma$** We note from Theorem 1 that finding the optimal choice of $\gamma$ is crucial. One way is to apply the efficient binary search during the certify process to find the optimal value of $\gamma$. Specifically, we set the initial value of $\gamma$ and compute the corresponding local Lipschitz constant $L_{f_L}^{(\gamma)}(x)$ at input $x$. By comparing the value between $\gamma$ and $R_{f_R}(f_L(x))/L_{f_L}^{(\gamma)}(x)$, we divide the search space into two halves at each iteration to narrow down the search space until $\gamma^* = R_{f_R}(f_L(x))/L_{f_L}^{(\gamma^*)}(x)$. Another way is to do a one-step search. Specifically, we first approximate the local Lipschitz constant $\tilde{L}_{f_L}^{(\gamma)}(x)$ at $x$ by averaging local Lipschitz constants of inference data given the inference $\gamma$. We then set $\gamma' = R_{f_R}(f_L(x))/\tilde{L}_{f_L}^{(\gamma)}(x)$ and re-calculate the local Lipschitz constant $\tilde{L}_{f_L}^{(\gamma')}(x)$ according to $\gamma'$. Finally, we compute the approximate optimal $\gamma^* = R_{f_R}(f_L(x))/\tilde{L}_{f_L}^{(\gamma')}(x)$. Overall, we show the certification process in detail in Algorithm 1.

**Remark 2: Split Optimization** In Theorem 1, we presented our result for an arbitrary split of the classifier. In principle, we can also optimize over how we split the classifier. If the base classifier is a composition of $K$ functions and the left part of the classifier $f_{L^{(s)}}$ contains $s$ layers and $f_R$ contains $(K - s)$ layers, then we can find the optimal split $s^*$ by varying $s$ from $0, 1, 2, \ldots, K$. We can observe that selecting $s = 0$ corresponds to conventional randomized smoothing whereas $s = K$ corresponds to label smoothing. In our experiments (see Section 4), we find that it is sufficient to split after a few layers (e.g., split the classifier after the $s = 1^{st}$ layer, $f_L = f_1$) and this alone suffices to outperform the state-of-art methods (Salman et al., 2019; Cohen et al., 2019; Jeong & Shin, 2020; Jeong et al., 2021) on the CIFAR-10 dataset, where we show the comparison of certified radius (RS vs *SPLITZ*) in Fig 3. Similar behavior can also be observed on other image datasets such as ImageNet and MNIST. We further discuss the impact of different splitting strategies in Section 4.

**Training Methodology for *SPLITZ*** In this Section, we present the details on training the *SPLITZ* classifier. The key to ensuring the certified robustness of the *SPLITZ* classifier is to keep the local Lipschitz constant of the left half of the classifier $f_L$ *small* while smoothing the right half of the classifier $f_R$. Let us denote $w_L, w_R$ as the training parameters of $f_L$ and $f_R$, respectively. We propose the following training loss function:

$$\min_{w_L, w_R} \frac{1 - \lambda}{N} \sum_{i=1}^{N} \mathbb{E}_\delta[\text{Loss}(f_R^{w_R}(f_L^{w_L}(x_i) + \delta), y_i)] + \frac{\lambda}{N} \sum_{i=1}^{N} \max(\theta, L_{f_L^{w_L}}^{(\gamma)}(x_i)), \quad (7)$$

where $\lambda \in [0, 1]$ is a hyperparameter controlling the tradeoff between the accuracy and robustness, $\theta$ is a learnable parameter to optimize the local Lipschitz constant, and $\text{Loss}(\cdot)$ is the loss function (e.g., cross entropy loss). Following the literature on randomized smoothing, we replace the expectation operators with their empirical estimates, and our loss function becomes:

$$\min_{w_L, w_R} \underbrace{\frac{1 - \lambda}{N} \sum_{i=1}^{N} \left( \frac{1}{Q} \sum_{q=1}^{Q} \text{Loss}(f_R^{w_R}(f_L^{w_L}(x_i) + \delta_q), y_i) \right)}_{\text{"Smoothing" loss for } f_R} + \underbrace{\frac{\lambda}{N} \sum_{i=1}^{N} \max(\theta, L_{f_L^{w_L}}^{(\gamma)}(x_i))}_{\text{Lipschitz regularization for } f_L}. \quad (8)$$

As illustrated in Eq 8, we first input the image $x_i$ to the left part of the classifier $f_L(x_i)$ and then add noise $\delta_q$, which forms the noisy samples $f_L(x_i) + \delta_q$. We then feed noisy samples to the right part of the classifier $f_R(f_L(x_i) + \delta_q)$ and obtain the corresponding prediction. Given the true label $y_i$, the loss (e.g., cross entropy) w.r.t $x_i$ can be computed. At the same time, the local Lipschitz constant of the left part of the classifier needs to be minimized. To this end, we propose a regularization term to the loss function, which controls the local Lipschitz constant of $f_L$. In addition, we do not want the value of the local Lipschitz constant to become too small during the training process, which may lead to a poor accuracy. Therefore, we set a learnable Lipschitz constant threshold $\theta$ for local Lipschitz constant of $f_L$, and use $\max(\theta, L_{f_L}^{(\gamma)}(x_i))$ as the regularization term.

**Computing the Local Lipschitz bound**: We note that both *SPLITZ* training as well as inference/certification require the computation of the local Lipschitz constant of the left half of the network, i.e., $f_L$. The simplest approach would be to use a bound on the global Lipschitz constant of $f_L$. For example, if $f_L$ is composed of $s$ layers, with each layer being a combination of an affine operation followed by ReLU nonlinearity, then the following simple bound could be used:

$$L_{f_L}^{(\gamma)}(x) \leq ||W_s||_2 \times ||W_{s-1}||_2 \ldots ||W_1||_2,$$

where $W_s$ is the weight matrix of layer $s$ and $||W_s||_2$ denotes the corresponding spectral norm. However, this bound, while easy to compute turns out to be quite loose. More importantly, it does not depend on the specific input $x$ as well as the parameter $\gamma$. Fortunately, bounding the local Lipschitz constant of a classifier is an important and a well studied problem. There are plenty of mechanisms to estimate the local Lipschitz bound of $f_L$. In principle, our *SPLITZ* classifier is compatible with these lo-

---

**Algorithm 1** *SPLITZ* Inference & Certification

1: Sample $n_0$ number of noise and augment $f_L(x)$: $f_L(x) + \delta_1, \ldots, f_L(x) + \delta_{n_0}$
2: Predict the top class from outputs of $f_R$: $c_A \leftarrow f_R(f_L(x) + \delta_1), \ldots, f_R(f_L(x) + \delta_{n_0})$
3: Sample $n$ number of noise and augment $f_L(x)$: $f_L(x) + \delta_1, \ldots, f_L(x) + \delta_n$
4: Approximate the lower confidence bound of the probability of the top class: $\underline{p_A} \leftarrow f_R(f_L(x) + \delta_1), \ldots, f_R(f_L(x) + \delta_n)$
5: **if** $\underline{p_A} > 0.5$ **then**
6:     Compute the certified radius of $f_R$: $R_{f_R}(f_L(x)) \leftarrow \sigma\Phi^{-1}(\underline{p_A})$
7:     Search the optimal $\gamma$ and calculate the corresponding local Lipschitz bound (Eq. 9).
8:     Compute the overall certified radius at $x$ (Theorem 1).
9:     **Return** prediction $c_A$ and robust radius $R_{g_{SPLITZ}}(x)$
10: **else**
11:     **Return** Abstain
12: **end if**

---

cal Lipschitz bound estimation algorithms. From the scope of this paper, we use the methodology proposed in (Huang et al., 2021) which leads to much tighter bounds on the local Lipschitz constant and maintain the specificity on the input $x$. Specifically, we apply the clipped version of activation layers (e.g. ReLU) to constrain each affine layer's output and obtain the corresponding upper bound (UB) and the lower bound (LB) for each affine layer, where the classifier is given an input $x$ around a $\gamma$ ball. We use an indicator function $I^v$ to represent index of the rows or columns in the weight matrices of each affine layer, which within the range from LB to UB. By multiplying each affine layer's weight matrix and each clipped activation layer' indicator matrix, the tighter local Lipschtz constant can be obtained. Assume $f_L$ network contains $s$-affine-layer neural network and each affine layer is followed by a clipped version of the activation layer, (upper bound of) the local Lipschitz constant $L$ of $f_L$ around the input $x$ is:

$$L_{f_L}^{(\gamma)}(x) \leq \| W_s I_{s-1}^v \|_2 \times \| I_{s-1}^v W_{s-1} I_{s-2}^v \|_2 \cdots \| I_1^v W_1 \|_2, \tag{9}$$

where $W_s$ is the weight matrix of layer $s$.

**Summary of *SPLITZ* Training Methodology** Overall, our training procedure is presented in Algorithm 2 (See Appendix B). During the process of computing local Lipschitz constant of $f_L$, for each iteration, we feed the input to the classifier $f_L$ and calculate the LB and UB of outputs of each affine layer in $f_L$ given the input $x$ within a $\gamma$ ball. We then can calculate the indicator matrix $I^v$ and compute the spectral norm of the reduced weight matrix $\| I_s^v W_s I_s^v \|$ for each layer $s$ in $f_L$ using *power iteration*. By multiplying the reduced weight matrix of each affine layer in $f_L$, we are able to arrive at the local Lipschitz constant of $f_L$. Secondly, we smooth the right half of the neural network $f_R$ by sampling from Gaussian noise with zero mean and adding it to the output of $f_L$. Then we feed the noisy samples $f_L(x) + \delta$ to $f_R$ and obtain the corresponding loss. Next, we minimize the overall loss and backward the parameters to optimize the overall network $f$. Finally, we certify the base classifier $f$ to obtain the Lipschitz smoothing classifier $g_{SPLITZ}$ as shown in Algorithm 1.

## 4 EVALUATION

In this section, we evaluate the *SPLITZ* classifier on three datasets, MNIST(LeCun et al., 1998), CIFAR-10 (Krizhevsky et al., 2009) and ImageNet (Russakovsky et al., 2015), where we demonstrate that our proposed approach consistently surpasses other state-of-the-art methods. For all datasets, we report the approximate certified test accuracy and certified radius of smoothed classifiers over test datasets (full test datasets in MNIST and CIFAR-10 datasets and a subsample of 1,000 test data in ImageNet dataset). Same as previous works, we vary the noise level $\sigma \in \{0.25, 0.5, 1.0\}$ for the smoothed models and certified the same noise level $\sigma$ during the inference time. To ensure a fair comparison with previous studies, we provide the highest reported results from each paper for the corresponding three levels of noise magnitudes. To improve certified robustness, we utilize the tighter local Lipschitz bound introduced in (Huang et al., 2021). For three datasets, we use the same model as previous works (Cohen et al., 2019; Carlini et al., 2023; Jeong et al., 2021; Jeong & Shin, 2020) (LeNet for MNIST, ResNet110 for CIFAR-10, ResNet50 for ImageNet). All the baseline mechanisms and more experimental details are described in Appendix E.

**Evaluation metric** Our evaluation metric to measure the certified robustness of the smooth classifier is based on the standard metric proposed in (Cohen et al., 2019): *the approximate certified test accuracy*, which can be estimated by the fraction of the test dataset which CERTIFY classifies are correctly classified and at the same time corresponding radius are larger than radius $\epsilon$ without abstaining. Another alternative metric is to measure the *average certified radius* (ACR) considered by (Zhai et al., 2020),which are provided in Appendix E. We show that *SPLITZ* consistently outperforms other mechanisms w.r.t ACR. For all experiments, we applied the $\ell_2$ norm input perturbation.

*SPLITZ* **Methodology** For all three datasets, we split the classifier after $1^{st}$ affine layer where the left half of the classifier contains one convolution layer followed by the clipped ReLU layer (See Appendix D). For the ImageNet dataset, the only difference is that we remove the *BatchNorm* layer after the $1^{st}$ affine layer and we replace the ReLU layer with the clipped ReLU layer in the first half of the network, which helps us obtain a tighter local Lipschitz bound of the first half of the classifier. The rest of the classifier is the same as original models (LeNet for MNIST, ResNet110 for CIFAR-10, ResNet50 for ImageNet).

**Dataset Configuration** For the MNIST, CIFAR-10 and ImageNet dataset(s), we draw $N = 10^5, 10^5$ and $10^4$ respectively noise samples to certify the smoothing model following (Cohen et al., 2019; Carlini et al., 2023; Jeong et al., 2021). We set the Lipschitz threshold (See Sec 3) as $\theta = 0.5, 0.5$ and $0.4$ respectively. For local Lipschitz constrained training, we set tradeoff parameter $\lambda$ (See Sec 3) evenly decrease from $0.7 - 0.3$, $1 - 0.7$ and $0.9 - 0.6$ respectively. We use one Nvidia P100 GPU to train the *SPLITZ* model with batch size 512, 256 and 128 respectively. We apply Adam Optimizer for three datasets. For the MNIST dataset, we train 150 epochs and set the initial learning rate as 0.1. The learning rate is decayed (multiplied by 0.1) by 0.01 at every 30 epochs (30th, 60th...). For the CIFAR-10 dataset, we train 400 epochs for the ResNet110 and set the initial learning rate as 0.001 and final learning rate as $10^{-6}$. The learning rate starts to evenly decay at each epoch from epoch 200. For the ImageNet dataset, we train 200 epochs for the ResNet50 and set the initial learning rate as 0.01. The learning rate starts to decay at each 40 epochs. We report our training and certifying time, along with more experimental details in Appendix E.

### 4.1 MAIN RESULTS

**Results on MNIST** As showed in Table 1, we can observe that *SPLITZ* outperforms other state-of-art approaches in almost every value of $\epsilon$. Impressively, we find that the *SPLITZ* classifier has a significant improvement when the value of $\epsilon$ is large. For instance, when $\epsilon = 2.50$, *SPLITZ* classifier achieves **88.2%** compared to state-of-art top-1 test accuracy 34.8% certified test accuracy on the MNIST dataset. Moreover, when we increase $\epsilon$ from 1.50 to 2.50, RS drops from 67.3% to 10.9% decreasing 56.4% test accuracy. *SPLITZ*, however, maintains higher certified test accuracy from 94.5% to 88.2% decreasing only 6.3% test accuracy.

**Results on CIFAR10** As shown in Table 2 and Fig 3, our method outperforms the state-of-art approaches for every value of $\epsilon$ on CIFAR-10 dataset. Interestingly, we find that the split Lipschitz training has a significant improvement when the value of $\epsilon$ is large. For instance, when $\epsilon = 1.0$, the model achieves **61.7%** top-1 test accuracy on CIFAR-10 dataset compared to state-of-art top-1 test accuracy 39.8%, an 61.7% improvement over the prior works. One hypothesis is that minimizing the Lipscitz bound of $f_1$ ($L \leq 1$) is able to boost the certified radius of the model. Intuitively, more sam-

| Method | Extra data | Certified accuracy at $\epsilon$ (%) | | | |
|---|---|---|---|---|---|
| | | 0.25 | 0.5 | 0.75 | 1.0 |
| PixelDP (Lecuyer et al., 2019) | ✗ | 22.0 | 2.0 | 0.0 | 0.0 |
| RS (Cohen et al., 2019) | ✗ | 61.0 | 43.0 | 32.0 | 22.0 |
| SmoothAdv (Salman et al., 2019) | ✗ | 67.4 | 57.6 | 47.8 | 38.3 |
| SmoothAdv (Salman et al., 2019) | ✓ | 74.9 | 63.4 | 51.9 | 39.6 |
| MACER (Zhai et al., 2020) | ✗ | 71.0 | 59.0 | 46.0 | 38.0 |
| Consistency (Jeong & Shin, 2020) | ✗ | 68.8 | 58.1 | 48.5 | 37.8 |
| SmoothMix (Jeong et al., 2021) | ✗ | 67.9 | 57.9 | 47.7 | 37.2 |
| Boosting (Horváth et al., 2021) | ✗ | 70.6 | 60.4 | 52.4 | 38.8 |
| DRT (Yang et al., 2021) | ✗ | 70.4 | 60.2 | 50.5 | 39.8 |
| ACES (Horváth et al., 2022) | ✗ | 69.0 | 57.2 | 47.0 | 37.8 |
| DDS (Carlini et al., 2023) | ✓ | 76.7 | 63.0 | 45.3 | 32.1 |
| DDS (finetuning) (Carlini et al., 2023) | ✓ | 79.3 | 65.5 | 48.7 | 35.5 |
| *SPLITZ* (ours) | ✗ | **81.2** | **75.8** | **69.4** | **61.7** |

Table 2: Comparison of the approximate certified test accuracy (%) on CIFAR-10 under $\ell_2$ norm perturbation. Extra data indicates whether their models incorporate other datasets in their models. Each entry lists the certified accuracy using numbers taken from respective papers. We set our values bold face when the value outperforms the best among the comparison methods.

| Method | Extra data | Certified accuracy at $\epsilon$ (%) | | | | |
|---|---|---|---|---|---|---|
| | | 0.5 | 1 | 1.5 | 2.0 | 3.0 |
| PixelDP (Lecuyer et al., 2019) | ✗ | 16.0 | 0.0 | 0.0 | 0.0 | 0.0 |
| RS (Cohen et al., 2019) | ✗ | 49.0 | 37.0 | 29.0 | 19.0 | 12.0 |
| SmoothAdv (Salman et al., 2019) | ✗ | 56.0 | 43.0 | 37.0 | 27.0 | 20.0 |
| MACER (Zhai et al., 2020) | ✗ | 57.0 | 43.0 | 31.0 | 25.0 | 14.0 |
| Consistency (Jeong & Shin, 2020) | ✗ | 50.0 | 44.0 | 34.0 | 24.0 | 17.0 |
| SmoothMix (Jeong et al., 2021) | ✗ | 50.0 | 43.0 | 38.0 | 26.0 | 20.0 |
| Boosting (Horváth et al., 2021) | ✗ | 57.0 | 44.6 | 38.4 | 28.6 | 21.2 |
| DRT (Yang et al., 2021) | ✗ | 46.8 | 44.4 | **39.8** | 30.4 | **23.2** |
| ACES (Horváth et al., 2022) | ✗ | 54.0 | 42.2 | 35.6 | 25.6 | 19.8 |
| DDS (Carlini et al., 2023) | ✓ | 71.1 | 54.3 | 38.1 | 29.5 | 13.1 |
| *SPLITZ* (ours) | ✗ | **58.4** | **46.2** | 38.2 | **31.6** | 20.2 |

Table 3: Comparison of the approximate certified test accuracy (%) on ImageNet under $\ell_2$ norm perturbation. We set our values bold face when the value outperforms the best among the comparison methods. Bold face with underline means that *SPLITZ* outperforms state-of-art methods except DDS (Carlini et al., 2023) with extra data. The columns and rows have the same meaning as in Table 2. ples are correctly classified while corresponding radius are larger than given $\epsilon$. In addition, we can observe the similar trend as MNIST dataset. *SPLITZ* maintains higher certified test accuracy (from 81.2% to 61.7%) when we increase $\epsilon$ from 0.25 to 1.00 compared to other state-of-art mechanisms.

**Results on ImageNet** We show the comparison of different certified robustness techniques on ImageNet dataset in Table 3. We observe similar trends to MNIST and CIFAR10 datasets, where *SPLITZ* is effective on certified robustness with a wide range of image datasets. It's noteworthy that *SPLITZ* outperforms other state-of-art mechanisms except DDS (Carlini et al., 2023), which uses denoising diffusion models with extra data. Moreover, *SPLITZ* consistently achieves better ACR (average certified radius) than other mechanisms on the ImageNet dataset as shown in Appendix E.

## 4.2 ABLATION STUDY

We also conduct an ablation study to explore the effects of hyperparameters in our proposed method on CIFAR-10 and MNIST datasets. We will explain the effect of global (local) Lipschitz bound, effect of input perturbation $\delta$ and effect of learnable Lipschitz threshold parameter $\theta$.

**Impact of splitting location** As mentioned in Section 3, our *SPLITZ* classifier can be optimized over different split ways, where we conduct the experiments and show our results in Table 4. For example, when $\epsilon = 2$, splitting after the $1^{st}$, or $2^{nd}$, or $3^{rd}$ layer result in certified accuracy of 91.7%, 74.7% and 1.3% respectively. These results indicate that splitting the neural network early achieves better performance. Intuitively, splitting the neural network early helps the model minimize the local Lipschitz bound, which improves the certified robustness leading to a higher certified test

| Location of Splitting | Certified Test Accuracy at $\epsilon$ (%) | | | | | | | | |
|---|---|---|---|---|---|---|---|---|---|
| | 0.50 | 0.75 | 1.00 | 1.25 | 1.50 | 1.75 | 2.00 | 2.25 | 2.50 |
| 1$^{st}$ affine layer | **97.4** | **96.9** | **96.2** | **95.4** | **94.5** | **93.0** | **91.7** | **90.1** | **88.2** |
| 2$^{nd}$ affine layer | 94.5 | 92.9 | 90.7 | 87.8 | 84.3 | 79.9 | 74.7 | 68.2 | 60.3 |
| 3$^{rd}$ affine layer | 92.6 | 87.7 | 80.5 | 67.2 | 32.3 | 7.8 | 1.3 | 0.0 | 0.0 |

Table 4: Comparison of certified test accuracy of *SPLITZ* with Gaussian noise $\sigma = 0.5$ for varying the splitting layer on MNIST dataset with LeNet.

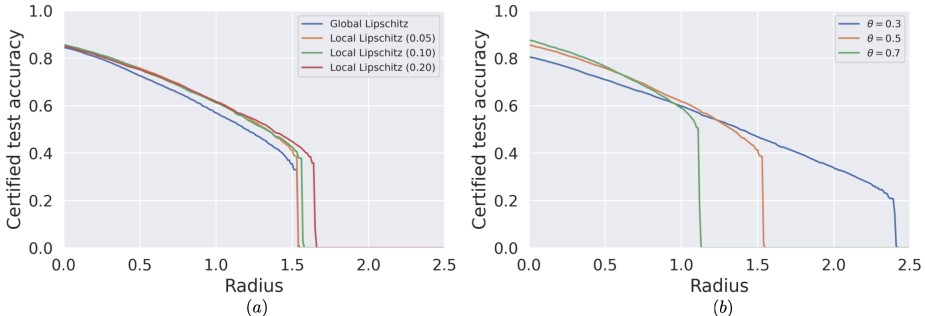

Figure 4: Comparison of certified test accuracy of *SPLITZ* with Gaussian noise $\sigma = 0.25$ for (a) global vs local Lipschitz bound ($\gamma$), (b) varying the local Lipschitz threshold $\theta \in \{0.3, 0.5, 0.7\}$.

accuracy given the same $\epsilon$. As the splitting becomes "deeper", estimating the local Lipshitz constant also becomes harder, which implies that a looser bound leads to smaller certified radius.

**Effect of global (local) Lipschitz constant of the first half of the classifier** As shown in Fig 4 (a), we investigate the effect of (upper bound of) the Lipschitz constant of left half of the classifier on certified test accuracy. Interestingly, we can observe that tighter Lipschitz bound gives better certified accuracy given the same radius. Furthermore, using a bound on the local Lipschitz constant to compute the certified accuracy is always better than using the global Lipschitz constant. This is also clearly evident from the result of Theorem 1.

**Effect of $\theta$ (Lipschitz threshold)** As shown in Fig 4(b), we analyze the effect of the training threshold $\theta$ (See Eq 8). For smaller values of $\epsilon$, *SPLITZ* with higher Lipschitz constant achieves better performances. Conversely, *SPLITZ* with a smaller Lipschitz constant can boost certified radius, which obtains a relative higher certified test accuracy when $\epsilon$ is larger. This ablation study further validate that the key of our *SPLITZ* classifier is to maintain a relative small Lipschitz constant for the left half of the classifier.

| | Certified Test Accuracy at $\epsilon$ (%) | | | | | |
|---|---|---|---|---|---|---|
| $\gamma$ | 0.00 | 0.25 | 0.50 | 0.75 | 1.00 | 1.25 |
| 0.05 | 85.6 | 81.2 | 75.8 | 69.4 | 61.7 | 53.2 |
| 0.10 | 85.7 | 81.3 | 75.2 | 68.5 | 61.2 | 52.4 |
| 0.20 | 85.3 | 80.5 | 75.2 | 69.0 | 61.6 | 54.2 |

Table 5: Comparison of certified test accuracy of *SPLITZ* with Gaussian noise $\sigma = 0.25$ for varying $\gamma$ (the size of the ball around input $x$) on CIFAR-10 dataset.

**Effect of $\gamma$ (size of radius around input $x$)** According to above results, constraining the local Lipschitz constant achieves better performance. To further explore the benefit of local Lipschitz constrained training, it is necessary to explore the indicator matrix $I^v$ in Eq 9, which depends on the size of the ball around the input (i.e., the hyperparameter $\gamma$). In Table 5, we show how varying $\gamma$ impacts the certified test accuracy for different values of $\epsilon$. We observe that smaller values of $\gamma$ lead to higher certified accuracy for all values of $\epsilon$.

## 5 DISCUSSION AND CONCLUSION

In this paper, we presented *SPLITZ*, a novel and practical certified defense mechanism, where we constrained the local Lipschitz bound of the left half of the classifier and smoothed the right half of the classifier with noise. Furthermore, we provide the comprehensive theoretical analysis of the certified robustness guarantee of *SPLITZ*. We showed results on several benchmark datasets and obtained significant improvements over state-of-art methods for MNIST, CIFAR-10 and close to state-of-the-art for ImageNet. We believe that combining the core idea of *SPLITZ* with other recent techniques, such as denoising diffusion models, adversarial re-training etc., can be a fruitful next step to further improve certified robustness.

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

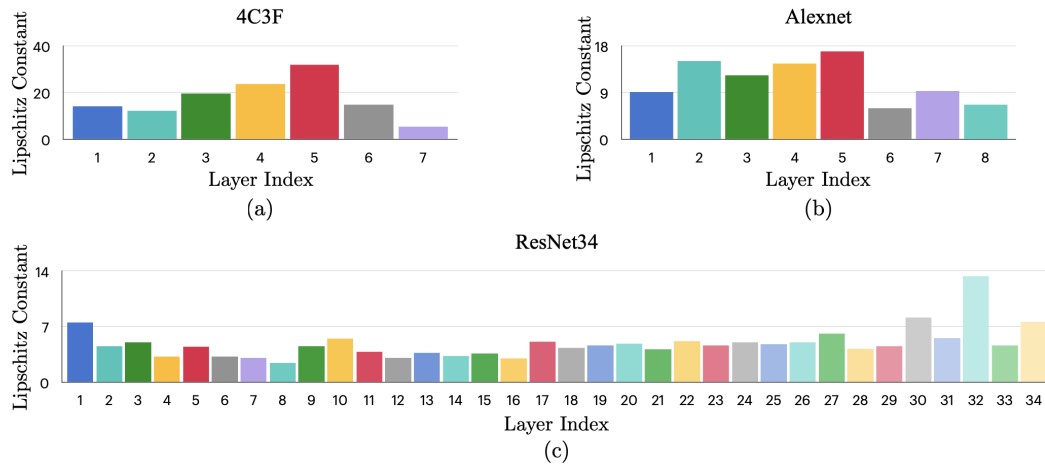

Figure 5: Lipschitz Constants of each affine layer in pretrained models:(a) 4C3F model (there are 4 convolution layers and 3 fully- connected layers in the neural network. ) (Huang et al., 2021) , (b) Alexnet model (Krizhevsky et al., 2012), (c) ResNet34 model (He et al., 2016). We can observe the similar trends that right half of the model usually contain a larger Lipschitz constant, while the left half of the model preserves a relatively smaller Lipschitz constant.

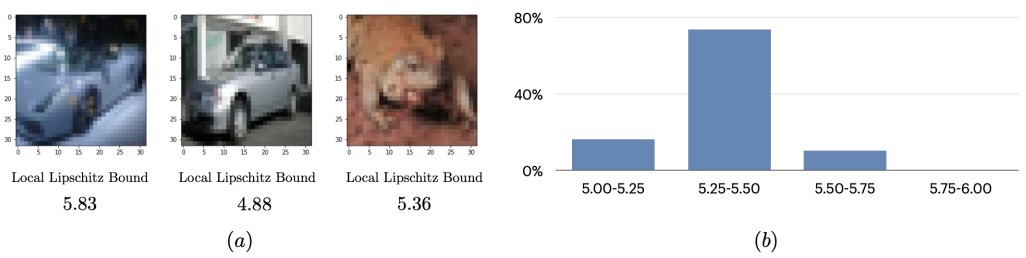

Figure 6: (a) Local Lipschitz bound for three random CIFAR-10 images on Alexnet, (b) Percentage analysis of local Lipschitz bound in CIFAR-10 test data.

## A    ADDITIONAL LIPSCHITZ BOUND RESULTS

In this Section, we provide additional Lipschitz constants results in the prevalent neural networks in Fig 5. We can observe the similar trends as previous that the right half of the neural network is more *unstable* than the right half of the neural network. As shown in Fig 6 (a), we notice considerable variation in the values of local Lipschitz constants across different input images, a trend that is consistent throughout the entire CIFAR-10 test dataset as depicted in Fig 6 (b). These findings lead us to reconsider the efficacy of directly smoothing the input. Such an approach doesn't cater to the observed heterogeneity. Alternatively, injecting noise at an intermediary step within the classifier can make the model more robust to disturbances

## B    COMPARISON AND EXTENSION OF *SPLITZ*

**Comparison to (Lecuyer et al., 2019)** Different from (Lecuyer et al., 2019), which estimates the sensitivity of the first half of the network and apply differential privacy mechanisms to preserve the certified robustness, we *minimize* the Lipschitz constant while applying the randomized smoothing to *improve* the certified robustness. We prove the certified robustness by the property of Lipschitz constant and randomized smoothing, and we derive the closed-form certified radius. We illustrate the schematic of our *SPLITZ* mechanism in Fig 2. Moreover, we show the training algorithm in Algorithm 2

---

**Algorithm 2** *SPLITZ* Training

---

**Input:** Training set $D = \{x_i, y_i\}_{i=1}^N$; noise level $\sigma$, training steps $T$, Lipschitz threshold $\theta$, training input perturbation $\gamma$

1: **for** $t = 0, \ldots, T-1$ **do**
2:      Compute local Lipschitz constant of $f_L$: $L(f_L, x, \gamma) \leftarrow cal\_local\_Lip(f_L, x, \gamma)$ .
3:      Sample noise $\delta$ and add it to outputs of $f_L$ to obtain noise samples: $f_L(x) + \delta$
4:      Feed the noise samples to $f_R$ network to get the corresponding predictions: $f_R(f_L(x) + \delta)$
5:      Set the local Lipschitz threshold $\theta$ and minimize the loss function in Eq. 8.
6: **end for**

**Function** $cal\_local\_Lip(f_L, x, \gamma)$

1: Compute the $UB_k$ and $LB_k$ for each layer $k$ in $f_L(x)$ given the perturbation $\gamma$ around input $x$
2: Compute the indicator matrix $I_k^v$ for each layer $k$
3: Compute the local Lipschitz constant $L_{f_L}^{(\gamma)}(x)$ (Eq. 9)

**Return** $L_{f_L}^{(\gamma)}(x)$

---

**Compatibility of *SPLITZ* with other defenses** In addition, we claim that the *SPLITZ* mechanism is also compatible with other RS based certified robust techniques, such as adversarial smoothing (Salman et al., 2019), mixsmoothing (Jeong et al., 2021) or denoising diffusion models (Carlini et al., 2023). As an example, (Carlini et al., 2023) propose a denosing mechanism using a diffusion model, which achieves the state-of-the-art. Our *SPLITZ* classifier contains two parts, left part is constrained by a small local Lipschitz constant while right part is smoothed by noise, which is same as a randomized smoothing based mechanism. Thus, our model can easily add a diffusion denoising model after the noise layer (after $f_L(x) + \delta$) and then feed the denoised samples into the $f_R$. Similarly, for adversarial smoothing or mixsmoothing, *SPLITZ* is adaptable to feed either adversarial examples ($f_L(x') + \delta$) or mixup samples ($f_L(\tilde{x}) + \delta$) respectively to the right half of the classifier $f_R$.

## C    PROOF OF THE THEOREM 1

**Theorem 1.** Let us denote $L_{f_L}^{(\gamma)}(x)$ as the $\gamma$-local Lipschitz constant of the function $f_L$ at $x$ in a ball of size $\gamma$, and $R_{f_R}(f_L(x))$ as the certified radius of the function $f_R$ at the input $f_L(x)$, with probability at least $(1-\alpha)$. Then, for any input $x$, with probability $1-\alpha$, $g_{SPLITZ}(x)$ has a certified radius of at least,

$$R_{g_{SPLITZ}}(x) = \max_{\gamma \geq 0} \ \min \left\{ \frac{R_{f_R}(f_L(x))}{L_{f_L}^{(\gamma)}(x)}, \gamma \right\} \tag{10}$$

*Proof.* Let us consider an input $x$ to *SPLITZ* classifier $g_{SPLITZ}(\cdot)$ and define the following function

$$\tilde{g}(u) \triangleq \underset{c \in \mathcal{Y}}{argmax} \ \mathbb{P}_\delta(f_R(u + \delta) = c). \tag{11}$$

We first note from Definition 3 that $g_{SPLITZ}(x)$ can be written as $g_{SPLITZ}(x) = \tilde{g}(f_L(x))$, where the function $\tilde{g}$ is the smoothed version of $f_R$. We are given that the smooth version $\tilde{g}$ has a certified radius of $R_{\tilde{g}}(u) \triangleq R_{f_R}(u)$ with probability at least $1-\alpha$. This is equivalent to the statement that for all $u'$ such that $||u - u'||_p \leq R_{f_R}(u)$, we have $\tilde{g}(u) = \tilde{g}(u')$. We are also given $L_{f_L}^{(\gamma)}(x)$, the $\gamma$-local Lipschitz constant of the function $f_L$ at $x$ in a ball of size $\gamma$. This implies that for all $||x - x'||_p \leq \gamma$,

$$||f_L(x) - f_L(x')||_p \leq L_{f_L}^{(\gamma)}(x)||x - x'||_p \tag{12}$$

Now, setting $u = f_L(x)$ and $u' = f_L(x')$, we obtain

$$||u - u'||_p = ||f_L(x) - f_L(x')||_p \leq L_{f_L}^{(\gamma)}(x)||x - x'||_p. \tag{13}$$

Now observe that ensuring $g_{SPLITZ}(x) = g_{SPLITZ}(x')$ is equivalent to ensuring $\tilde{g}(f_L(x)) = \tilde{g}(f_L(x'))$, which in turn is equivalent to $\tilde{g}(u) = \tilde{g}(u')$. Thus, if we ensure that

$$L_{f_L}^{(\gamma)}(x)||x - x'||_p \leq R_{f_R}(f_L(x)) \leftrightarrow ||x - x'||_p \leq \frac{R_{f_R}(f_L(x))}{L_{f_L}^{(\gamma)}(x)} \tag{14}$$

then $g_{SPLITZ}(x) = g_{SPLITZ}(x')$. However, we also note that $||x - x'||_p \leq \gamma$, therefore the certified radius is given by $\min(R_{f_R}(f_L(x))/L_{f_L}^{(\gamma)}(x), \gamma)$. We finally note that the choice of $\gamma$ (size of the ball) was arbitrary, and we can pick the *optimum* choice that yields the largest radius. This leads to the final expression for certified radius for *SPLITZ*:

$$R_{g_{SPLITZ}}(x) = \max_{\gamma \geq 0} \ \min \left\{ \frac{R_{f_R}(f_L(x))}{L_{f_L}^{(\gamma)}(x)}, \gamma \right\} \tag{15}$$

and completes the proof of the Theorem. $\qquad\square$

## D  LIPSCHITZ CONSTRAINED TRAINING

From the scope of this paper, we utilize the local Lipschitz contrained training for the left half of the classifier introduced in (Huang et al., 2021). We focus on $l_2$ norm denoted as $\|\cdot\|$. Now we consider a neural network $f$ containing $L$ affine layers (parameterized by $w$) each followed by a clipped version ReLU$\theta$, which is defined as follows:

$$ReLU\theta(x) = \begin{cases} 0, & \text{if } x \leq 0 \\ x, & \text{if } 0 < x < \theta \\ \theta, & \text{if } x \geq \theta \end{cases} \tag{16}$$

The neural network maps input $x$ to output $f(x)$ using the following architecture:

$$z_1 = x; z_l(x) = ReLU\theta(W_l x), z_{L+1} = W_L z_L \tag{17}$$

We define the perturbation around the input $x$ as:

$$x' = x + \epsilon, \| \epsilon \| \leq \delta, \delta \geq 0 \tag{18}$$

By adding perturbation around input $x$ within a $\delta$ ball, $z(x')$ can be bounded element-wise as $LB \leq z(x') \leq UB$, where $LB$ and $UB$ can obtain by bound propagation methods (Gowal et al., 2018; Lee et al., 2020). We then define the diagonal matrix $I^v$ to represent the entries where ReLU$\theta$'s outputs are *varying*:

$$I^v(i, i) = \begin{cases} 1, & \text{if } UB_i > 0 \ and \ LB_i < \theta \\ 0, & \text{otherwise} \end{cases} \tag{19}$$

Next, the output of the ReLU$\theta$ $D^v$ can be defined as follows:

$$D^v(i, i) = \begin{cases} \mathbb{1}(ReLU\theta(z_l^i) > 0), & \text{if } I^v(i, i) = 1 \\ 0, & \text{otherwise} \end{cases} \tag{20}$$

where $\mathbb{1}$ denote the indicator function. Then the local Lipschitz bound at input $x$ is:

$$L_{local}(x, f) \leq \| W_L I_{L-1}^v \| \| I_{L-1}^v W_{L-1} I_{L-2}^v \| \cdots \| I_1^v W_1 \| \tag{21}$$

As stated in (Huang et al., 2021), it straight forward to prove $_{L-1}^v W_{L-1} I_{L-2}^v \| \leq \| W_{L-1} \|$ using the property of eigenvalues. We briefly prove it following from Huang et al. (2021).

*Proof.* Let $W' = [W \ I]^T$, The singular value of $W'$ is defined as the square roots of the eigenvalues of $W'^T W'$. We know the following

$$W'^T W' = W^T W + I^T I \geq W^T W. \tag{22}$$

Therefore, we get the following result:

$$\| W' \| \geq \| W \| \tag{23}$$

We complete the proof. $\qquad\square$

Next, we will give a toy example to further illustrate the idea of local Lipschitz bound.

**A toy example** Here we provide a similar toy example as mentioned in (Huang et al., 2021). Consider a 2-layer neural network with ReLU$\theta$ activation layer:

$$x \to \text{Linear1}(W^1) \to \text{ReLU}\theta \to \text{Linear2}(W^2) \to y \tag{24}$$

where $x \in \mathcal{R}^3$ and $y \in \mathcal{R}$ and $W^l$ denotes the weight matrix for layer $l$. Moreover the threshold $\theta = 1$.

Given the input [1,-1,0] with $\ell_2$ perturbation 0.1. Assume the weight matrices are:

$$W^1 = \begin{bmatrix} 2 & 0 & 0 \\ 0 & 2 & 0 \\ 0 & 0 & 1 \end{bmatrix}, W^2 = [1, 1, 1] \tag{25}$$

Thus, we have the following:

$$\text{Input } [1, -1, 0] \to \begin{bmatrix} [0.9 & 1.1] \\ [-1.1 & -0.9] \\ [-0.1 & 0.1] \end{bmatrix} \times \begin{bmatrix} 2 & 0 & 0 \\ 0 & 2 & 0 \\ 0 & 0 & 1 \end{bmatrix} \to \begin{bmatrix} [1.8 & 2.2] \\ [-2.2 & -1.8] \\ [-0.1 & 0.1] \end{bmatrix} \tag{26}$$

According to the above upper bound (UB) and lower bound (LB), we obtain the $I_V$ function as follows:

$$I_V^1 = \begin{bmatrix} 0 & 0 & 0 \\ 0 & 0 & 0 \\ 0 & 0 & 1 \end{bmatrix} \tag{27}$$

Overall, we have the local Lipschitz bound as follows:

$$L_{local}(x, f) \leq \parallel W^2 I_V^1 \parallel \parallel I_V^1 W^1 \parallel = 1 \tag{28}$$

For the global Lipschitz bound, we have the following:

$$L_{global} \leq \parallel W^2 \parallel \parallel W^1 \parallel = 4 \tag{29}$$

Overall, we can find that the local Lipschiz bound is much tighter than the global Lipschitz bound.

## E    ADDITIONAL EXPERIMENTAL RESULTS

In this section, we provide additional results for the three datasets, e.g., MNIST, CIFAR-10, ImageNet. We first provide the details of three datasets. Next, we illustrate the baselines used in our paper. Note that we report the numbers (certified test accuracy, average certified radius) from respective papers. In the main results part, we report our certified test accuracy for each value of noise $\sigma$. Furthermore, we provide additional ablation study results to further investigate the performance of the *SPLITZ* classifier.

### E.1    TRAINING DETAILS

For all value of $\sigma$, we keep the value of training $\sigma$ and testing $\sigma$ to be the same. We apply the noise samples $n_0 = 100$ to predict the most probably class $c_A$ and denote $\alpha = 0.001$ as the confidence during the certifying process. Furthermore, we use $n = 100,000, n = 100,000, n = 10,000$ to calculate the lower bound of the probability $p_A$ for the MNIST. CIFAR-10 and ImageNet respectively. Moreover, to maintain a relatively small local Lipschitz constant of left half of the *SPLITZ* classifier, we set the threshold of clipped ReLU (see Sec D) as 1 for all three datasets. For estimating the local Lipschitz constant of the left half of the classifier, the power iteration is 5, 2, 2 during the training for MNIST, CIFAR-10 and ImageNet respectively following from (Huang et al., 2021).

| Datasets | Architecture | # of GPUs | Training per epoch (s) | Certifying per image (s) |
|----------|--------------|-----------|------------------------|--------------------------|
| MNIST | LeNet | 1 | 15.1 | 1.0 |
| CIFAR-10 | ResNet110 | 1 | 59.5 | 50.7 |
| ImageNet | ResNet50 | 1 | 4810.5 | 83.1 |

Table 6: *SPLITZ* training time and certifying time across three datasets (MNIST using LeNet, CIFAR-10 using ResNet 110 and ImageNet using ResNet 50).

### E.2 DETAILS OF DATASETS

**MNIST** dataset (LeCun et al., 1998) contains handwritten digits usually used for image classification problems. The dataset is comprised of a total of 70,000 images, with 60,000 images in the training set and 10,000 in the test set. The dataset has 10 classes and is in grayscale format. We pre-process the MNIST dataset using normalization.

**CIFAR-10** (Krizhevsky et al., 2009) dataset consists of 60,000 RGB images distributed across 10 categories: airplane, automobile, bird, cat, deer, dog, frog, horse, ship, and truck. Each category is represented by 6,000 images. The dataset is divided into a training set with 50,000 images and a test set containing 10,000 images. We employ the conventional data augmentation techniques of random horizontal flipping and random translation by up to 4 pixels, consistent with methods used in other baseline studies (Carlini et al., 2023; Cohen et al., 2019; Jeong et al., 2021). Additionally, we normalize each image on a pixel-by-pixel basis.

**ImageNet** (Russakovsky et al., 2015) dataset contains over 1.2 million training images and 50,000 validation images, labeled to 1,000 classes. For the purpose of data augmentation, we apply random cropping at a 224x224 resolution along with random resizing and horizontal flips to the training images. During testing, however, we execute a 224x224 center crop after resizing the images to a 256x256 size.

### E.3 BASELINE MECHANISMS

We compare our method with various existing techniques proposed for robust training of smoothed classifiers, as listed below: (a) PixelDP (Lecuyer et al., 2019): cerified robust training with differential privacy mechanism; (b) RS (Cohen et al., 2019): standard randomized smoothing with the classifier trained with Gaussian augmentation; (c) SmoothAdv (Salman et al., 2019): adversarial training combined with randomized smoothing; (d) MACER (Zhai et al., 2020): a regularization approach which maximizes the approximate certified radius; (e) Consistency (Jeong & Shin, 2020): a KL-divergence based regularization that minimizes the variance of smoothed classifiers $f(x + \delta)$ across $\delta$; (f) SmoothMix (Jeong et al., 2021): training on convex combinations of samples and corresponding adversarial on smoothed classifier; (g) Boosting (Horváth et al., 2021): a soft-ensemble scheme on smooth training; (h) DRT (Yang et al., 2021): a lightweight regularized training on robust ensemble ML models; (i) ACES (Horváth et al., 2022): a selection-mechanism combined with a smoothed classifier; (j) DDS (Carlini et al., 2023): a denoised diffusion mechanism combined with a smoothed classifier.

### E.4 TRAINING AND CERTIFYING TIME

Our *SPLITZ* model needs to vary the value of $\gamma$ (the size of the ball around input $x$) during the training epoch. Thus, we may need relatively more time to obtain the optimal model. To solve this, we apply the early stop mechanism to obtain the better optimized model during the training process. At the same time, we decay our learning rates during the training process. We use one Nvidia P100 GPU to train our *SPLITZ* model and report our training time (certifying time) for each one epoch (one image) in the Table 6.

| $\sigma$ | Methods | MNIST | CIFAR-10 | ImageNet |
|---|---|---|---|---|
| | RS* | 0.911 | 0.424 | - |
| | SmoothAdv* | 0.932 | 0.544 | - |
| | MACER | 0.918 | 0.556 | - |
| 0.25 | Consistency | 0.928 | 0.552 | - |
| | SmoothMix | 0.933 | 0.548 | - |
| | SPLITZ | **1.664** | **1.025** | - |
| | RS* | 1.553 | 0.525 | 0.733 |
| | SmoothAdv* | 1.687 | 0.684 | 0.825 |
| | MACER | 1.583 | 0.726 | 0.831 |
| 0.50 | Consistency | 1.697 | 0.726 | 0.822 |
| | SmoothMix | 1.694 | 0.737 | 0.846 |
| | SPLITZ | **3.412** | **1.562** | **0.974** |
| | RS* | 1.620 | 0.542 | 0.875 |
| | SmoothAdv | 1.779 | 0.660 | 1.040 |
| | MACER | 1.520 | 0.792 | 1.008 |
| 1.00 | Consistency | 1.819 | 0.816 | 0.982 |
| | SmoothMix | 1.823 | 0.773 | 1.047 |
| | SPLITZ | **2.886** | **1.970** | **1.320** |

Table 7: Comparison of average certified radius (ACR) across three different datasets (MNIST, CIFAR-10, ImageNet). We can observe that for three datasets, *SPLITZ* consistently achieves better results compared to other state-of-art mechanisms. * is reported by (Jeong & Shin, 2020; Jeong et al., 2021)

### E.5 RESULTS ON ACR

In this section, we investigate the performance of *SPLITZ* using average certified radius (ACR), where we measure the correct samples' average certified radius over the test datasets (MNIST, CIFAR-10 and ImageNet). As shown in Table 7, we provide the comprehensive comparison results of average certified radius (ACR) compared to other certified robust techniques. Our *SPLITZ* consistently outperforms others, where the ACR of *SPLITZ* is almost twice that of others' certified radius on MNIST dataset. For instance, when $\sigma = 0.5$, the ACR of *SPLITZ* is 3.412, where the state-of-the-art is 1.697.

### E.6 MAIN RESULTS

### E.6.1 MNIST RESULTS

For the results of MNIST dataset as shown in Table 8, we observe that our *SPLITZ* classifier consistently obtains better test certified accuracy while we have a large value of $\epsilon$ (e.g., especially when $\epsilon$ is larger than 1.5). Moreover, we found that when noise level is 0.5, our model achieves best results. The interesting phenomenon observed by us is that *SPLITZ* model with noise 0.5 maintains a relatively small local Lipschitz bound for the left half of the classifier (average $L_{f_L}^{(\gamma)}$ is 0.49). While *SPLITZ* model with noise 0.25 obtain a relatively larger local Lipschitz bound, where average $L_{f_L}^{(\gamma)}$ is 0.52. We believe that this is the main reason that our *SPLITZ* model with noise 0.5 achieves better results.

Another phenomenon we observed is that when the noise level becomes larger, our *SPLITZ* model can be sensitive to noise, which may need to search different layers (See Remark 2) to obtain the better results. For instance, our *SPLITZ* model can barely learn the information from noisy samples while splitting after the $1^{st}$ affine layer when $\sigma = 1$. Thus, by searching the splitting location among

| | | Certified Test Accuracy at $\epsilon$ (%) | | | | | | | | | | |
|---|---|---|---|---|---|---|---|---|---|---|---|
| $\sigma$ | Methods | 0.00 | 0.25 | 0.50 | 0.75 | 1.00 | 1.25 | 1.50 | 1.75 | 2.00 | 2.25 | 2.50 |
| 0.25 | RS* | 99.2 | 98.5 | 96.7 | 93.3 | 0.0 | 0.0 | 0.0 | 0.0 | 0.0 | 0.0 | 0.0 |
| | MACER | 99.0 | 99.0 | 97.0 | 95.0 | 0.0 | 0.0 | 0.0 | 0.0 | 0.0 | 0.0 | 0.0 |
| | Consistency | 99.5 | 98.9 | 98.0 | 96.0 | 0.0 | 0.0 | 0.0 | 0.0 | 0.0 | 0.0 | 0.0 |
| | SmoothMix | 99.5 | 99.0 | 98.2 | 97.0 | 0.0 | 0.0 | 0.0 | 0.0 | 0.0 | 0.0 | 0.0 |
| | DRT | 99.5 | 98.6 | 97.6 | 96.7 | 0.0 | 0.0 | 0.0 | 0.0 | 0.0 | 0.0 | 0.0 |
| | SPLITZ | 98.2 | 97.7 | 97.0 | 96.2 | **95.2** | **94.0** | **92.3** | **49.9** | 0.0 | 0.0 | 0.0 |
| 0.50 | RS* | 99.2 | 98.3 | 96.8 | 94.3 | 89.7 | 81.9 | 67.3 | 43.6 | 0.0 | 0.0 | 0.0 |
| | MACER | 99.0 | 98.0 | 96.0 | 94.0 | 90.0 | 83.0 | 73.0 | 50.0 | 0.0 | 0.0 | 0.0 |
| | Consistency | 99.2 | 98.6 | 97.6 | 95.9 | 93.0 | 87.8 | 78.5 | 60.5 | 0.0 | 0.0 | 0.0 |
| | SmoothMix | 99.0 | 98.4 | 97.4 | 95.7 | 93.0 | 88.5 | 81.8 | 70.7 | 0.0 | 0.0 | 0.0 |
| | DRT | 99.2 | 98.6 | 97.4 | 95.6 | 93.3 | 88.5 | 81.2 | 68.6 | 0.0 | 0.0 | 0.0 |
| | SPLITZ | 98.3 | 97.9 | 97.4 | **96.9** | **96.2** | **95.4** | **94.5** | **93.0** | **91.7** | **90.1** | **88.2** |
| 1.00 | RS* | 96.3 | 94.4 | 91.4 | 86.8 | 79.8 | 70.9 | 59.4 | 46.2 | 32.5 | 19.7 | 10.9 |
| | MACER | 89.0 | 85.0 | 79.0 | 75.0 | 69.0 | 61.0 | 54.0 | 45.0 | 36.0 | 28.0 | - |
| | Consistency | 95.0 | 93.0 | 89.7 | 85.4 | 79.7 | 72.7 | 63.6 | 53.0 | 41.7 | 30.8 | 20.3 |
| | SmoothMix | 95.5 | 93.5 | 90.5 | 86.2 | 80.6 | 73.4 | 64.3 | 54.5 | 44.9 | 37.1 | 29.3 |
| | DRT | 96.0 | 94.1 | 90.2 | 86.6 | 80.7 | 73.2 | 63.7 | 54.3 | 46.7 | 40.3 | 34.7 |
| | SPLITZ | 93.7 | 92.3 | 90.6 | **88.4** | **85.8** | **83.1** | **80.0** | **76.7** | **72.7** | **68.4** | **63.2** |

Table 8: Comparison of certified test accuracy of *SPLITZ* with Gaussian noise $\sigma = [0.25, 0.5, 1]$ on MNIST dataset. * is reported by (Jeong & Shin, 2020; Jeong et al., 2021).

| | | Certified Test Accuracy at $\epsilon$ (%) | | | | | | | | | |
|---|---|---|---|---|---|---|---|---|---|---|---|
| $\sigma$ | Methods | 0.00 | 0.25 | 0.50 | 0.75 | 1.00 | 1.25 | 1.50 | 1.75 | 2.00 | 2.25 |
| 0.25 | RS* | 75.0 | 60.0 | 43.0 | 26.0 | 0.0 | 0.0 | 0.0 | 0.0 | 0.0 | 0.0 |
| | SmoothAdv* | 74.0 | 67.0 | 57.0 | 47.0 | 0.0 | 0.0 | 0.0 | 0.0 | 0.0 | 0.0 |
| | MACER | 81.0 | 71.0 | 59.0 | 43.0 | 0.0 | 0.0 | 0.0 | 0.0 | 0.0 | 0.0 |
| | Consistency | 77.8 | 68.8 | 58.1 | 48.5 | 0.0 | 0.0 | 0.0 | 0.0 | 0.0 | 0.0 |
| | SmoothMix | 77.1 | 67.9 | 57.9 | 47.7 | 0.0 | 0.0 | 0.0 | 0.0 | 0.0 | 0.0 |
| | Boosting | 83.4 | 70.6 | 60.4 | 52.4 | 0.0 | 0.0 | 0.0 | 0.0 | 0.0 | 0.0 |
| | DRT | 81.5 | 70.4 | 60.2 | 50.5 | 0.0 | 0.0 | 0.0 | 0.0 | 0.0 | 0.0 |
| | SPLITZ | **85.6** | **81.2** | **75.8** | **69.4** | **61.7** | **53.2** | **41.4** | 0.0 | 0.0 | 0.0 |
| 0.5 | RS* | 65.0 | 54.0 | 41.0 | 32.0 | 23.0 | 15.0 | 9.0 | 4.0 | 0.0 | 0.0 |
| | SmoothAdv* | 50.0 | 46.0 | 44.0 | 40.0 | 38.0 | 33.0 | 29.0 | 23.0 | 0.0 | 0.0 |
| | MACER | 66.0 | 60.0 | 53.0 | 46.0 | 38.0 | 29.0 | 19.0 | 12.0 | 0.0 | 0.0 |
| | Consistency | 64.3 | 57.5 | 50.6 | 43.2 | 37.8 | 33.9 | 29.9 | 25.2 | 0.0 | 0.0 |
| | SmoothMix | 65.0 | 56.7 | 49.5 | 43.3 | 37.2 | 31.7 | 25.7 | 19.8 | 0.0 | 0.0 |
| | Boosting | 69.0 | 60.4 | 49.8 | 44.8 | 38.8 | 34.4 | 30.4 | 25.0 | 0.0 | 0.0 |
| | DRT | 69.7 | 61.2 | 50.9 | 44.4 | 39.8 | 36.0 | 30.4 | 24.1 | 0.0 | 0.0 |
| | SPLITZ | **78.2** | **75.1** | **71.2** | **66.8** | **62.4** | **58.2** | **53.7** | **49.1** | **44.9** | **40.2** |
| 1 | RS* | 47.0 | 39.0 | 34.0 | 28.0 | 21.0 | 17.0 | 14.0 | 8.0 | 5.0 | 3.0 |
| | SmoothAdv* | 45.0 | 41.0 | 38.0 | 35.0 | 32.0 | 28.0 | 25.0 | 22.0 | 19.0 | 17.0 |
| | MACER | 45.0 | 41.0 | 38.0 | 35.0 | 32.0 | 29.0 | 25.0 | 22.0 | 18.0 | 16.0 |
| | Consistency | 48.1 | 43.9 | 39.3 | 34.7 | 30.0 | 27.6 | 24.7 | 22.0 | 19.5 | 17.3 |
| | SmoothMix | 47.1 | 42.5 | 37.5 | 33.8 | 30.2 | 26.7 | 23.4 | 20.2 | 17.2 | 14.7 |
| | Boosting | 49.6 | 44.0 | 38.2 | 35.6 | 32.6 | 29.2 | 25.8 | 22.0 | 19.8 | 16.2 |
| | DRT | 50.4 | 44.4 | 40.8 | 37.0 | 34.2 | 30.1 | 26.8 | 23.9 | 20.3 | - |
| | SPLITZ | **66.5** | **63.7** | **60.9** | **57.6** | **54.7** | **51.7** | **48.5** | **45.3** | **42.5** | **39.6** |

Table 9: Comparison of certified test accuracy of *SPLITZ* with Gaussian noise $\sigma = [0.25, 0.5, 1]$ on CIFAR-10 dataset. * is reported by (Zhai et al., 2020). In this table, we only report the mechanisms, which provides the certified accuracy respectively for noise $\sigma \in \{0.25, 0.5, 1\}$.

different layers, we split after the $2^{n}d$ affine layers and obtain the corresponding better results in Table 8.

| $\sigma$ | Methods | Certified Test Accuracy at $\epsilon$ (%) | | | | | | |
|---|---|---|---|---|---|---|---|---|
| | | 0.00 | 0.50 | 1.00 | 1.50 | 2.00 | 2.50 | 3.00 |
| | RS* | 57.0 | 46.0 | 37.0 | 29.0 | 0.0 | 0.0 | 0.0 |
| | SmoothAdv* | 54.0 | 49.0 | 43.0 | 37.0 | 0.0 | 0.0 | 0.0 |
| | MACER | 64.0 | 53.0 | 43.0 | 31.0 | 0.0 | 0.0 | 0.0 |
| 0.5 | Consistency | 55.0 | 50.0 | 44.0 | 34.0 | 0.0 | 0.0 | 0.0 |
| | SmoothMix | 55.0 | 50.0 | 43.0 | 38.0 | 0.0 | 0.0 | 0.0 |
| | Boosting | 58.6 | 52.0 | 44.6 | 38.4 | 0.0 | 0.0 | 0.0 |
| | SPLITZ | **65.2** | **58.4** | **46.2** | 33.4 | **22.6** | 0.0 | 0.0 |
| | RS* | 44.0 | 38.0 | 33.0 | 26.0 | 19.0 | 15.0 | 12.0 |
| | SmoothAdv* | 40.0 | 37.0 | 34.0 | 30.0 | 27.0 | 25.0 | 20.0 |
| | MACER | 48.0 | 43.0 | 36.0 | 30.0 | 25.0 | 18.0 | 14.0 |
| 1 | Consistency | 41.0 | 37.0 | 32.0 | 28.0 | 24.0 | 21.0 | 17.0 |
| | SmoothMix | 40.0 | 37.0 | 34.0 | 30.0 | 26.0 | 24.0 | 20.0 |
| | Boosting | 45.0 | 41.0 | 37.2 | 34.0 | 28.6 | 24.6 | 21.2 |
| | SPLITZ | **57.4** | **52.8** | **45.2** | **38.2** | **31.6** | **26.6** | 20.2 |

Table 10: Comparison of certified test accuracy of *SPLITZ* with Gaussian noise $\sigma = [0.5, 1]$ on ImageNet dataset. Similar as above, we only report the mechanisms, which provides the certified accuracy respectively for noise $\sigma \in \{0.5, 1\}$.

### E.6.2 CIFAR-10 RESULTS

As shown in Table 9, our method outperforms the state-of-art approaches in every value of $\epsilon$ on CIFAR-10 dataset. In addition, we find that the *SPLITZ* has a significant improvement when the value of $\epsilon$ is large. For instance, when $\epsilon = 1.0$, the model achieves **61.7%** top-1 test accuracy on CIFAR-10 dataset compared to state-of-art top-1 test accuracy 39.8%, an 55.0% improvement over the prior works. *SPLITZ* maintains higher certified test accuracy (from 81.2% to 61.7%) when we increase $\epsilon$ from 0.25 to 1.00 compared to other state-of-art mechanisms (e.g. RS drops from 61.0% to 22.0 %).

### E.6.3 IMAGENET RESULTS

As shown in Table 10, our method outperforms the state-of-art approaches in most of the time on ImageNet dataset. Since the key of *SPLITZ* is that we need to maintain relatively small local Lipschitz bound for our left half of the classifier. Compared to MNIST and CIFAR-10, ImageNet is a harder dataset with a more complex model architecture, which makes local Lipschitz bound for our left half of the classifier looser and hard to minimize during the training process, which provides space to improve our model.

### E.7 ADDITIONAL ABLATION STUDY

We provide additional results while varying the training threshold $\theta$ (See Sec 3 for more details) on CIFAR-10 dataset as shown in Fig 7 and Table 11. This experiment further proves that with a smaller local Lipschitz constant of left half of the classifier, our *SPLITZ* classifier can boost the certified test accuracy when $\epsilon$ is larger. Conversely, when local Lipschitz bound of left half of the classifier is larger, we have higher certified test accuracy for smaller $\epsilon$ as shown in Fig 7.

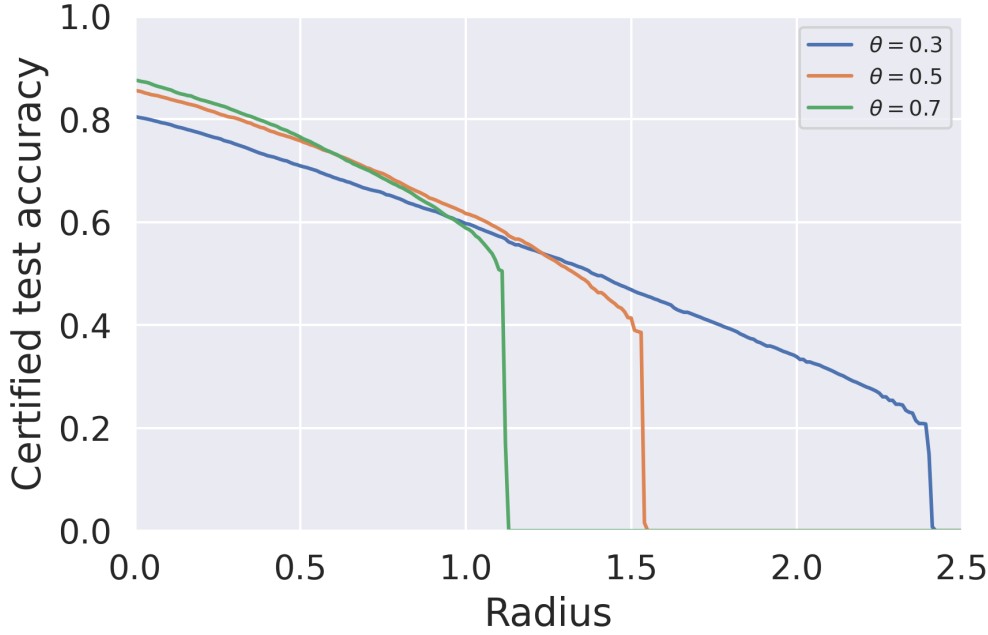

Figure 7: Varying the local Lipschitz threshold $\theta \in \{0.3, 0.5, 0.7\}$ during the training process.

| Training Threshold $\theta$ | Certified Test Accuracy at $\epsilon$ (%) | | | | | | | | |
|---|---|---|---|---|---|---|---|---|---|
| | 0.00 | 0.25 | 0.50 | 0.75 | 1.00 | 1.25 | 1.50 | 1.75 | 2.00 |
| 0.3 | 80.6 | 76.3 | 70.9 | 65.7 | 59.7 | 53.5 | 46.9 | 40.5 | 33.8 |
| 0.5 | 85.6 | 81.2 | 75.8 | 69.4 | 61.7 | 53.2 | 41.4 | 0.0 | 0.0 |
| 0.7 | 87.3 | 82.9 | 76.5 | 68.5 | 58.9 | 0.0 | 0.0 | 0.0 | 0.0 |

Table 11: Comparison of certified test accuracy of *SPLITZ* with Gaussian noise $\sigma = 0.25$ for varying the local Lipschitz constant training threshold $\theta$ on CIFAR-10 dataset.

