# OpenReview forum: "SPLITZ: Certifiable Robustness via Split Lipschitz Randomized Smoothing"
_ICLR.cc/2024/Conference — ICLR 2024 Conference Withdrawn Submission_

### Official Review · Reviewer_CJku · 2023-10-25

**Soundness:** 1 poor
**Presentation:** 2 fair
**Contribution:** 1 poor
**Rating:** 3
**Confidence:** 5

**Summary:**

This paper proposes SPLITZ, a new approach for training neural networks with certifiable robustness guarantees by exploiting both Lipschitz continuity and randomized smoothing (RS). The key idea is to split the neural network into two parts, the first part would have constrained Lipschitz constants, while the latter layers are smoothed by randomized smoothing. The authors claim that this new method has the advantages of both certified approaches (Lipschitz continuity and RS) in a single framework.  The authors provide a theoretical analysis to compute the robustness radius for SPLITZ classifiers and demonstrate the effectiveness of SPLITZ with experiments on MNIST, CIFAR, and ImageNet.

**Strengths:**

This idea is interesting, and given that Lipschitz continuity and randomized smoothing are two well-established methods for certifiable robustness, it is interesting to work toward a unification of the two approaches.

**Weaknesses:**

**Major weakness: I believe the paper to be flawed**

- The results on CIFAR10 show an increase of 21.9 points of certified robustness for eps = 1 with the L2 norm compared to the state of the art. This increase is extremely high and, in my opinion, suspicious. I took the time to check the code and it seems that the authors normalize the inputs of the model. The authors mention this in Appendix E.2 DETAILS OF DATASETS.

After a review of the code, it seems that the authors use a function called `get_architecture` to initialize the model, this function is defined as:

```
def get_architecture(arch: str, dataset: str) -> torch.nn.Module:
     """ Return a neural network (with random weights)

    :param arch: the architecture - should be in the ARCHITECTURES list above
    :param dataset: the dataset - should be in the datasets.DATASETS list
    :return: a Pytorch module
    """
    if arch == "resnet50" and dataset == "imagenet":
        model = resnet50(pretrained=False).cuda()
    elif arch == "cifar_resnet20":
        model = resnet_cifar(depth=20, num_classes=10).cuda()
    elif arch == "cifar_resnet110":
        model = resnet_cifar(depth=110, num_classes=10).cuda()
    elif arch == "mnist_lenet":
        model = lenet().cuda()
    normalize_layer = get_normalize_layer(dataset)
    return torch.nn.Sequential(normalize_layer, model)
```
 and the `get_normalize_layer` function is:

```
def get_normalize_layer(dataset: str) -> torch.nn.Module:
    """Return the dataset's normalization layer"""
    if dataset == "imagenet":
        return NormalizeLayer(_IMAGENET_MEAN, _IMAGENET_STDDEV)
    elif dataset == "cifar10":
        return NormalizeLayer(_CIFAR10_MEAN, _CIFAR10_STDDEV)
    elif dataset == "mnist":
        return NormalizeLayer(_MNIST_MEAN, _MNIST_STDDEV)

_IMAGENET_MEAN = [0.485, 0.456, 0.406]
_IMAGENET_STDDEV = [0.229, 0.224, 0.225]

_CIFAR10_MEAN = [0.4914, 0.4822, 0.4465]
_CIFAR10_STDDEV = [0.2023, 0.1994, 0.2010]

_MNIST_MEAN = [0.1307]
_MNIST_STDDEV = [0.3081]
```

If that's the case, the certified radius should be scaled accordingly, for example, if CIFAR10 images are scaled with
```
_CIFAR10_MEAN = [0.4914, 0.4822, 0.4465]
```
then if I'm not mistaken, the certified radius should be divided by `1/min(_CIFAR10_STDDEV) ≈ 5.0`. This would explain the very high certified robustness of CIFAR10.

- Regarding the results on ImagetNet, the results are not better but on par with the state-of-the-art DDS approach (Carlini et al. 2023). This again is surprising given that the authors use a simple ResNet50 while Carlini et al. 2023 have used a combined model (diffusion + ViT classifier BEiT large model) with +800M parameters. Again, the normalization problem could explain this discrepancy.

**Other comments:**
- The proposed approach has already been investigated in the preprint [1].
- The authors do not seem to be aware of a large body of work on Lipschitz networks. The sentence "Lipschitz constrained training is often only feasible for smaller (few layers) neural networks" is false. The work of Meunier et al. [2] successfully trained a 1000-layer Lipschitz network by constraining all layers to have a Lipschitz equal to 1. Furthermore, there are many papers that the authors did not acknowledge, the sentence: "The main challenge is that accurate estimation of Lipschitz constants becomes infeasible for larger networks, and upper bounds become loose, leading to empty bounds on the certified radius." is true, but 1-Lipshitz networks actually solve this problem see [2, 3, 4, 5, 6].
- Using MNIST to demonstrate an adversarial robustness approach is not convincing to me due to the nature of the dataset (toy dataset).

[1] Zeng et al. Certified Defense via Latent Space Randomized Smoothing with Orthogonal Encoders
[2] Meunier et al., A Dynamical System Perspective for Lipschitz Neural Networks ICML 2022
[3] Araujo et al., A Unified Algebraic Perspective on Lipschitz Neural Networks, ICLR 2023
[4] Trockman et al., Orthogonalizing Convolutional Layers with the Cayley Transform, ICLR 2021
[5] Singla et al, Skew Orthogonal Convolutions, ICML 2021
[6] Prach et al., Almost-Orthogonal Layers for Efficient General-Purpose Lipschitz Networks, ECCV 2022

**Questions:**

Can the authors comment on the normalization issue?

---

> ### Author Response · Authors · 2023-11-18
> **Response to Reviewer CJku (1/2)**
>
> 1. **Can author comment on the normalization process?**\
> Thank you for your insightful and constructive feedback. Your comments on the normalization process in our work are highly appreciated and have prompted a deeper examination of our methodology.\
> Initially, we adopted normalization in the first layer to align our approach with existing baseline methodologies, ensuring a fair comparison of results. However, your observations led us to recognize potential issues with this approach. In response, we have conducted a series of experiments to better understand the impact of normalization on our model's performance.\
> Our initial experiment involved removing the normalization layer, which, interestingly, resulted in a decrease in certified accuracy under the same training parameters. This finding prompted further experimentation. We then tested the effect of applying normalization post-noise insertion, which yielded similar performance outcomes. Finally, we incorporated the normalization layer in the left half of our classifier while focusing on minimizing the local Lipschitz constant of the left half of the classifier. This was achieved by adjusting the loss function to include the multiplied maximum variance among the input channels and the previous local Lipschitz constant. These modifications led to notable findings. Our revised SPLITZ model not only addressed the initial concerns but also demonstrated superior performance compared to other baseline models. We show the certified test accuracy w.r.t $\epsilon$  in the Table 1 as follows \
> $\begin{array}{|c|c|c|c|}
> \hline
> Method & \epsilon = 1.50 & \epsilon =1.75 & \epsilon =2.00 & \epsilon =2.25 & \epsilon =2.50 \\\\ \hline
> RS [1] & 67.3 & 46.2 & 32.5 & 19.7 & 10.9 \\\\ \hline
> MACER [2] & 73.0 & 50.0 & 36.0 & 28.0 & -\\\\ \hline
> Consistency [3] &82.2 & 70.5 & 45.5 & 37.2 & 28.0 \\\\ \hline
> SmoothMix [4]  & 81.8 & 70.7 & 44.9 & 37.1 & 29.3 \\\\ \hline
> DRT [5] &83.3 & 69.6 & 48.3 & 40.3 & 34.8 \\\\ \hline
> SPLITZ (revised)  & 83.1 & \textbf{75.9} & \textbf{66.2} & \textbf{54.3} & \textbf{40.5}\\\\ \hline
> \end{array}$\
> As shown in the above table, we achieve the comparative results when $\epsilon = 1.50$. Our SPLITZ classifier outperforms other baselines when $\epsilon$ is larger. However, we acknowledge that further optimization of training parameters may be required for larger datasets, such as CIFAR-10 and ImageNet.\
> We plan to include a detailed account of these experiments and their implications in the appendix of our revised paper, with a summarized version in the main text. Your feedback has been instrumental in guiding these improvements, and we are grateful for your thorough and thoughtful review.
> 2. **Can you explain the difference between the preprint [6] and yours work?**\
> Thanks for bringing this preprint [6] to our attention. While this preprint [6] also proposes the idea of breaking the network into two halves, there are several key distinctions as we elaborate upon next. The preprint [6] uses an orthogonal convolutional network-based encoder followed by a randomized smoothing classifier. The key idea herein is to ensure that the global Lipschitz constant of the first half of the network (termed as "encoder" in the preprint) is close to $1$, then, the certified radius is essentially given by the ceritified radius of the second half (obtained by randomized smoothing).\
> Our work introduces a novel approach where we focus on measuring the $\gamma$-local Lipschitz constant of the network's first half. This is a strategic choice as it enables the capture of local information specific to each input $x$, offering a more nuanced understanding of the network’s behavior at an individual input level. To further enhance certified robustness, we have integrated the local Lipschitz constant of the network's first half into the loss function as a regularization term. This methodology aims to maintain a smaller local Lipschitz constant (specifically, $L_{f_L}^{(\gamma)} < 1$) for the first half of the network, which in turn significantly boosts the certified radius. For example, in the CIFAR-10 datasets, we found that the average local Lipschitz constant for the first half of the network is approximately 0.62, even though the global Lipschitz constants could be significantly higher. Furthermore, in Theorem $1$, we show that the certified radius can then be further optimized by varying $\gamma$ around each individual input $x$. Specifically, our certified radius is given by $\underset{\gamma \geq 0 }{\max}~~\min \left\\{\frac{R_{f_\text{R}}(f_\text{L}(x))}{L_{f_\text{L}}^{(\gamma)}(x)}, \gamma \right\\}$.
> This aspect of our work represents a unique and substantial deviation from previous studies, focusing on localized sensitivity and its impact on network robustness.

---

> ### Author Response · Authors · 2023-11-18
> **Response to Reviewer CJku (2/2)**
>
> 3. **Why not include more works in the Lipschitz constrained training?**\
> Thank you for directing us to the literature on Lipschitz-constrained training. We will certainly enhance our paper with additional discussion on this topic. Our proposed framework is notably adaptable. As mentioned in our comment on "Room for Improvement" and Appendix B, we discuss the integration of methods that a) improve local Lipschitz constants training in the first half of the network, and b) augment the second half with techniques like denoising models.
> 4. **Why do you use MNIST dataset?**\
> We acknowledge that compared to datasets like CIFAR-10 and ImageNet, MNIST is relatively smaller in scale. However, there are compelling reasons for its inclusion in our study. Firstly, MNIST is widely recognized as a benchmark dataset in the realm of randomized smoothing frameworks, as highlighted in various research papers [2,3,4,5]. Therefore, incorporating MNIST alongside CIFAR-10 and ImageNet allows for a more comprehensive and equitable comparison across standard datasets.\
> Secondly, a retrospective analysis of the state-of-the-art certified accuracy achievements on MNIST reveals there is still a significant room for improvement, particularly at higher values of $\epsilon$ (i.e., the budget of adversarial perturbations). For example, when $\epsilon$ is set to $2$, the current best-certified accuracy stands at only 48.3\% as proposed by [5]. This indicates a substantial opportunity for advancements using our methodology, particularly in enhancing certified accuracy under conditions of large $\epsilon$. From our revised experiments (accounting for the normalization as you pointed out), we achieve a certified test accuracy of $66.2\%$.  Our study aims to address this gap, demonstrating the effectiveness of our approach in optimizing certified robustness in a well-established and challenging benchmark scenario.
>
> Best regards,
>
> The authors
>
> [1] J. Cohen, E. Rosenfeld, and Z. Kolter, “Certified adversarial robustness via randomized smoothing,” in international conference on machine learning, pp. 1310–1320, PMLR, 2019.\
> [2] R. Zhai, C. Dan, D. He, H. Zhang, B. Gong, P. Ravikumar, C.-J. Hsieh, and L. Wang, “Macer: Attack-free and scalable robust training via maximizing certified radius,” in International Conference on Learning Representations, 2020.\
> [3] J. Jeong and J. Shin, “Consistency regularization for certified robustness of smoothed classifiers,” Advances in Neural Information Processing Systems, vol. 33, pp. 10558–10570, 2020.\
> [4] J. Jeong, S. Park, M. Kim, H.-C. Lee, D.-G. Kim, and J. Shin, “Smoothmix: Training confidence-calibrated smoothed classifiers for certified robustness,” Advances in Neural Information Processing Systems, vol. 34, pp. 30153–30168, 2021.\
> [5] Z. Yang, L. Li, X. Xu, B. Kailkhura, T. Xie, and B. Li, “On the certified robustness for ensemble models and beyond,” arXiv preprint
> arXiv:2107.10873, 2021.\
> [6] H. Zeng, J. Su, and F. Huang, “Certified defense via latent space randomized smoothing with orthogonal encoders,” arXiv preprint arXiv:2108.00491, 2021.

---

> > ### Comment · Reviewer_CJku · 2023-11-18
> > **Response to Rebuttal**
> >
> > Thank you for your detailed rebuttal.
> > Could you upload a revision of the code?

---

> > > ### Author Response · Authors · 2023-11-19
> > > **Response to Reviewer CJku**
> > >
> > > Thank you for your reply. Please find the revised code here: https://anonymous.4open.science/r/SPLITZ-DDC9.
> > >
> > > Best regards,
> > >
> > > The authors

---

> > > > ### Comment · Reviewer_CJku · 2023-11-22
> > > > **Response to Rebuttal**
> > > >
> > > > Thank you for the detailed rebuttal.
> > > >
> > > > In light of the new results on MNIST, the performance of the approach has been significantly reduced (e.g., for $\epsilon = 2.5$ the certified accuracy went from 88.2% to 40.5%). While the proposed approach seems to be competitive for MNIST (toy dataset), it is clearly not certain that it will generalize to more complex datasets.
> > > >
> > > > We can consider all (or most) of the experiments in the current version of the paper to be incorrect. I suggest that the authors re-run all the experiments with the correct normalization for all the datasets they have considered to demonstrate the competitiveness of the proposed method.
> > > >
> > > > Since the main flaw of the paper has been fixed, I'll increase my score from 1 to 3, but I still recommend rejection at this point.

---

### Official Review · Reviewer_3vvt · 2023-10-29

**Soundness:** 3 good
**Presentation:** 3 good
**Contribution:** 4 excellent
**Rating:** 8
**Confidence:** 3

**Summary:**

The authors present a novel algorithm to obtain certified robustness with high probability (in the sense of randomized smoothing), named SPLITZ. By combining local Lipschitz regularization in the first layer(s) with randomized smoothing in the latent space, the authors improve on the state-of-the-art by a sizeable margin on MNIST and CIFAR-10.

**Strengths:**

The idea behind the proposed approach is novel and conceptually simple/intuitive: to the best of my knowledge, this is the first work combining Lipschitz-based certified training schemes with randomized smoothing.
The paper is mostly well-written, with a clear presentation of the required technical background (sometimes in the appendix) and of the main technical building blocks of SPLITZ.
What stands out the most, though, is the experimental section, showing that SPLITZ outperforms previous approaches (even those using additional data) by a significant margin on both MNIST and CIFAR-10, with the performance improvement increasing with the perturbation radius.

**Weaknesses:**

To my mind, the main weakness of the work lies in the introduction of a fair number of hyper-parameters (train-time $\gamma$, $\theta$, $\lambda$), which will inevitably increase the overall runtime overhead of the proposed approach. Analogously, it would be nice to see a detailed analysis of the overhead incurred by the optimization over $\gamma$ (remark 1).

In addition, I think that the presentation itself could be somewhat improved in a couple of instances. For instance, the authors repeatedly state that certified training is either randomized smoothing or Lipschitz-based methods, somewhat ignoring the family of methods that train against network relaxations (for instance, IBP, CROWN-IBP, and more recent works such as SABR, TAPS, CC/MTL-IBP, etc.): these methods do not have an explicit Lipschitz estimation. Analogously, it is claimed that a large certified radius is equivalent to a small Lipschitz constant, but such is a sufficient rather than necessary condition for robustness. There is a formatting issue in page 8, where the body of the text and the caption become hard to visually separate.
Furthermore, some sections are not entirely self-contained: for instance, the paragraph before equation (9) does not explain how LB/UBs are contained, and defers to the appendix important details of the Lipschitz part of the training.

**Questions:**

Could the authors provide an indication of the overhead of running SPLITZ (including the hyper-parameter tuning process) with respect to the reported baselines?

The authors refer to $\theta$ as a "learnable" parameter: do you mean tunable hyper-parameter? If so, why does not $\lambda$ suffice?

Judging from "dataset configuration", it would feel like SPLITZ is quite sensitive to the $\lambda$ schedule. How was this tuned?

Results on MNIST and CIFAR-10 are remarkable. However, how do the authors explain the fact that performance improvements (if any) upon the baselines are significantly smaller on ImageNet?

It would be interesting to hear the authors' opinion as to why the effect of the splitting location on performance increases witht $\sigma$.

---

> ### Author Response · Authors · 2023-11-18
> **Response to Reviewer 3vvt (1/2)**
>
> 1. **Could the authors provide an indication of the overhead of running SPLITZ (including the hyper-parameter tuning process) with respect to the reported baselines?**\
> Thank you for your insightful query. The essence of our SPLITZ methodology lies in training the network's left half to have a relatively small local Lipschitz constant, while apply the randomized smoothing on the right half of the classifier. To achieve a small local Lipschitz constant for the first half, we initially adopt the settings from [1], setting $\lambda$ between 1 and 0.7. At the same time, we set the threshold of the local Lipschitz constant $\theta$ to be 0.5 by default. Adjustments to $\lambda$ or $\theta$ are made based on test accuracy outcomes. For the right half of the neural network which applies the randomized smoothing (RS), the overhead of running is comparable as RS [2].\
> For the optimization over the $\gamma$ happening in the inference time (certification process), as we mentioned in Remark 1, for affine layers, it is sufficient to do a one step search to obtain the $\gamma$. For non-affine layers, it is better to do the binary search over the $\gamma$ to accurately estimate the local Lipschitz constant. From our previous experimental results, it will usually take 10-15 iterations for each input to find the optimized $\gamma$, which is quite efficient compared to the overall training process.
> 2. **Is $\theta$ tunable? Why does not $\lambda$ suffice?**\
> Thank you for this excellent question. Indeed, the local Lipschitz threshold $\theta$ is tunable within our framework. The necessity of incorporating $\theta$ alongside the trade-off parameter $\lambda$ stems from the limitations observed when only $\lambda$ is used. Our experimental results reveal that solely constraining $\lambda$ tends to lead to an extremely small local Lipschitz constant, such as 0.01. While this might seem advantageous, it significantly compromises certified accuracy. Thus, by introducing $\theta$ as a tunable parameter, we establish a more balanced and effective control mechanism. This dual-parameter approach allows for a more nuanced optimization, ensuring that the reduction in the local Lipschitz constant does not excessively detract from the certified accuracy of the model.\
> Additionally, the inclusion of a local Lipschitz constant $\theta$ introduces a greater flexibility in managing a robustness budget. For instance, when presented with a predetermined budget for certified robustness, the ability to adjust $\theta$ becomes particularly advantageous. This flexibility enables a more strategic and efficient training approach, allowing us to maximize the certified robustness of the model within the constraints of the given budget. By tuning $\theta$, we can precisely calibrate the trade-off between robustness and other performance metrics, ensuring optimal model performance under specified robustness requirements.
> 3. **How was $\lambda$ tuned?**\
> In tuning $\lambda$, we initially established its weight range, guided by the settings referenced in [1], which spans from 1 to 0.7. This range serves as our starting point. We then closely monitored the corresponding certified test accuracy as a key performance indicator. Depending on the observed performance of the classifier, we either narrowed or maintained this range of $\lambda$. This iterative approach allowed us to fine-tune $\lambda$ with precision, ensuring that it optimally contributes to the balance between robustness and accuracy. The decision to adjust the range is made based on a careful analysis of the trade-offs between the strength of regularization (imposed by $\lambda$) and the resultant model performance.

---

> ### Author Response · Authors · 2023-11-18
> **Response to Reviewer 3vvt (2/2)**
>
> 4. **How do the authors explain the fact that performance improvements (if any) upon the baselines are significantly smaller on ImageNet?**\
> Thank you for raising this important question. The disparity in performance improvements between smaller datasets like MNIST and CIFAR-10 and a more complex dataset like ImageNet can be attributed to the challenges in minimizing the local Lipschitz constant for the network's left half. In simpler datasets, due to their limited complexity and diversity, it's relatively easier to achieve a lower local Lipschitz constant, which directly contributes to better performance enhancements. However, ImageNet, with its vast variety and complexity, presents a more challenging environment for this optimization process.\
> Our experimental findings support this observation. For example, under a noise variance of 0.25 on MNIST, CIFAR-10 and 0.5 on ImageNet, we observed that the local Lipschitz constants for MNIST, CIFAR-10, and ImageNet were 0.58, 0.62 and 0.80, respectively. This progression indicates an increasing difficulty in minimizing the Lipschitz constant as dataset complexity rises. Consequently, the performance improvements on ImageNet are not as pronounced as those on simpler datasets. This insight highlights the relationship between dataset complexity and the effectiveness of our method in optimizing the local Lipschitz constant for enhanced performance. Furthermore, to improve the performance of SPLITZ in ImageNet dataset, we can natural apply the randomized smoothing based mechanisms on the second half of the neural networks, such as adversarial smoothing [3], mixsmoothing [4] or denoising diffusion models [5] as we discussed in Appendix B.
> 5. **Why the effect of the splitting location on performance increases with $\sigma$?**\
> Thank you for your question. While varying the splitting location across the layers, to have a fair comparison, we apply the same noise in the model architecture. We will explore the splitting location with the effect of increasing $\sigma$ in our revised version, which will help us to have a better understanding of the effect of the splitting location with respect to $\sigma$.
>
> Best regards,
>
> The authors
>
> [1] Y. Huang, H. Zhang, Y. Shi, J. Z. Kolter, and A. Anandkumar, “Training certifiably robust neural networks with efficient local lipschitz bounds,” Advances in Neural Information Processing Systems, vol. 34, pp. 22745–22757, 2021.\
> [2] J. Cohen, E. Rosenfeld, and Z. Kolter, “Certified adversarial robustness via randomized smoothing,” in international conference on machine learning, pp. 1310–1320, PMLR, 2019.\
> [3] H. Salman, J. Li, I. Razenshteyn, P. Zhang, H. Zhang, S. Bubeck, and G. Yang, “Provably robust deep learning via adversarially trained smoothed classifiers,” Advances in Neural Information Processing Systems, vol. 32, 2019.\
> [4] J. Jeong, S. Park, M. Kim, H.-C. Lee, D.-G. Kim, and J. Shin, “Smoothmix: Training confidence-calibrated smoothed classifiers for certified robustness,” Advances in Neural Information Processing Systems, vol. 34, pp. 30153–30168, 2021.\
> [5] N. Carlini, F. Tramer, K. D. Dvijotham, L. Rice, M. Sun, and J. Z. Kolter, “(certified!!) adversarial robustness for free!,” in The Eleventh International Conference on Learning Representations, 2023.\

---

### Official Review · Reviewer_jHmH · 2023-10-31

**Soundness:** 3 good
**Presentation:** 3 good
**Contribution:** 3 good
**Rating:** 8
**Confidence:** 3

**Summary:**

This paper introduces a novel technique for achieving certified adversarial robustness by combining the principles of Lipschitz bounded networks with randomized smoothing. The approach involves partitioning a neural network into two components, where the first is bounded by a local Lipschitz constraint, and the second is robustified through randomized smoothing. The authors present a training procedure designed to reinforce the respective parts—ensuring Lipschitz continuity in the former and noise resilience in the latter. This allows the model to outperform state-of-the-art L2 certificates for image classification.

The proposed method has been evaluated on image datasets MNIST, CIFAR-10, and ImageNet. It consistently outperforms the state of the art on MNIST and CIFAR-10 datasets. However, on the ImageNet dataset, it only outperforms the state-of-the-art methods that do not use additional data for some of the certified radii. It does not quite outperform the method that uses additional data.

**Strengths:**

1. The ideas in the paper are articulated with clarity and are easy to follow.
2. The paper proposes a novel technique that combines two established methods with considerable efficacy.
3. Along with theoretical robustness guarantees, it proposes a training procedure to optimize the robustness criteria (Lipschitzness for the first part and robustness to random noise for the second) needed for this method.
4. It outperforms state-of-the-art techniques of certified robustness for MNIST and CIFAR-10 Image classification datasets.

**Weaknesses:**

1. The method does not consistently outperform existing approaches for ImageNet. Specifically, it does not perform as well as DDS, which leverages additional data to improve robustness.

Despite this, the improvement for smaller datasets is noteworthy. Keeping this in mind, I am leaning toward accepting this paper.

**Questions:**

Could this technique be adapted to incorporate additional training data (like DDS) to improve its performance on large-scale datasets like ImageNet?

---

> ### Author Response · Authors · 2023-11-18
> **Response to Reviewer jHmH**
>
> **Could this technique be adapted to incorporate additional training data (like DDS) to improve its performance on large-scale datasets like ImageNet?**\
> Thanks for your reviews and this is a great question! Indeed, our proposed mechanisms can incorporate other mechanisms into two ways: first, we can adopt more tight Lipschitz constrained training to achieve a relatively small local Lipschitz constant of the first half of the network; secondly, we can natural apply the randomized smoothing based mechanisms on the second half of the neural networks, such as adversarial smoothing [1], mixsmoothing [2] or denoising diffusion models [3]. For instance, as we stated in Appendix B, Carlini et al. [3] proposes a denosing mechanism using a diffusion model, which achieves the state-of-the-art. Our SPLITZ classifier contains two parts, left part is constrained by a small local Lipschitz constant while right part is smoothed by noise, which is same as a randomized smoothing based mechanism. Thus, our model can easily add a diffusion denoising model after the noise layer (after $f_L(x)+\delta$) and then feed the denoised samples into the $f_{R}$. Similarly, for adversarial smoothing or mixsmoothing, SPLITZ is adaptable to feed either adversarial examples ($f_L(x')+\delta$) or mixup samples ($f_L(\tilde x)+\delta$) respectively to the right half of the classifier $f_R$.
>
> Best regards,
>
> The Authors
>
> [1] H. Salman, J. Li, I. Razenshteyn, P. Zhang, H. Zhang, S. Bubeck, and G. Yang, “Provably robust deep learning via adversarially trained smoothed classifiers,” Advances in Neural Information Processing Systems, vol. 32, 2019.\
> [2] J. Jeong, S. Park, M. Kim, H.-C. Lee, D.-G. Kim, and J. Shin, “Smoothmix: Training confidence-calibrated smoothed classifiers for certified robustness,” Advances in Neural Information Processing Systems, vol. 34, pp. 30153–30168, 2021.\
> [3] N. Carlini, F. Tramer, K. D. Dvijotham, L. Rice, M. Sun, and J. Z. Kolter, “(certified!!) adversarial robustness for free!,” in The Eleventh International Conference on Learning Representations, 2023.

---

> > ### Comment · Reviewer_jHmH · 2023-11-23
> > **Reviewer Comment**
> >
> > I would like to thank the authors for answering my question. This paper studies a novel and interesting approach for certifying the robustness of neural networks. The method performs well on smaller image datasets like MNIST and CIFAR-10 and could potentially be improved to perform well for larger datasets like ImageNet.
> >
> > Overall, I think the paper makes a positive contribution, justifying its acceptance.

---

### Official Review · Reviewer_jeFQ · 2023-11-06

**Soundness:** 2 fair
**Presentation:** 3 good
**Contribution:** 3 good
**Rating:** 5
**Confidence:** 3

**Summary:**

This paper combines Lipschitz networks with randomized smoothing (RS) to develop the SPLITZ method which splits a classifier into two halves, constrain the Lipschitz constant of the first half, and smooth the second half via randomization. The motivation is that many standard deep networks exhibit heterogeneity in Lipschitz constants across layers, and the proposed method is capable of exploiting this heterogeneity while improving scalability of RS. Training methods and related robustness theory are developed. Numerical results are presented to show that the proposed method achieves good results on MNIST, CIFAR-10 and ImageNet datasets.

**Strengths:**

1. The proposed method is more scalable than RS.

2. The idea of exploiting heterogeneity in Lipschitz constants across layers is interesting.

3. Numerical study is quite comprehensive.

**Weaknesses:**

1. The idea of combining Lipschitz networks and RS may not be that original. In general, Lipschitz training is not the only way to restrict the Lipschitz constant of networks. One can enforce various network structures so a prescribed Lipschitz constant is ensured. The following paper which appeared in 2021 combines orthogonal Lipschitz layers with RS:

Huimin Zeng, Jiahao Su, and Furong Huang. Certified defense via latent space randomized smoothing with orthogonal encoders. arXiv2021.

Therefore, on the conceptual level, the paper may not be that novel. Btw, the above paper should be cited.

2. The advantage of the proposed method over DDS is not that convincing. I mean, from Table 3, it seems that DDS is better for smaller perturbations?

3. When talking about constraining the Lipschitz constant of networks, the authors mainly focus on Lipschitz training and ignore a large body of works that use prescribed network structures to constrain the Lipschitz constants. The following list of papers is relevant and should be discussed:

Takeru Miyato, Toshiki Kataoka, Masanori Koyama, and Yuichi Yoshida. Spectral normalization for generative adversarial networks. ICLR, 2018.

Qiyang Li, Saminul Haque, Cem Anil, James Lucas, Roger B Grosse, and Joern-Henrik Jacobsen. Preventing gradient attenuation in lipschitz constrained convolutional networks. NeurIPS, 2019.

Asher Trockman and J Zico Kolter. Orthogonalizing convolutional layers with the cayley transform. ICLR, 2021

Sahil Singla and Soheil Feizi. Skew orthogonal convolutions. ICML, 2021.

Tan Yu, Jun Li, Yunfeng Cai, and Ping Li. Constructing orthogonal convolutions in an explicit manner. ICLR 2022.

Laurent Meunier, Blaise Delattre, Alexandre Araujo, and Alexandre Allauzen. A dynamical system perspective for lipschitz neural networks. ICML 2022.

Bernd Prach and Christoph H Lampert. Almost-orthogonal layers for efficient general-purpose lipschitz networks. ECCV 2022.

Xiaojun Xu, Linyi Li, and Bo Li. Lot: Layer-wise orthogonal training on improving l2 certified robustness. NeurIPS 2022.

Alexandre Araujo, Aaron Havens, Blaise Delattre, Alexandre Allauzen, and Bin Hu. A unified algebraic perspective on lipschitz neural networks. ICLR 2023.

Ruigang Wang, and Ian Manchester. Direct parameterization of lipschitz-bounded deep networks. ICML 2023.

**A big question:Why do not use the above Lipschitz networks for the first half and then integrate RS with the second half? How to compare such a network parameterization approach with the proposed method?**

**Questions:**

1. The idea of combining Lipschitz networks and RS may not be that new. Can the authors be more specific about the unique conceptual novelty of this paper?

2.  From Table 3, it seems that DDS is better for smaller perturbations? Is it possible to make the proposed method achieve better results than DDS even for small perturbations?

3. As mentioned above, there is a large body of literature on parameterizing networks in certain ways to enforce Lipschitz constraints. Why do not use the above Lipschitz networks for the first half and then integrate RS with the second half? How to compare such a network parameterization approach with the proposed method?

---

> ### Author Response · Authors · 2023-11-18
> **Response to Reviewer jeFQ**
>
> 1. **Can the authors be more specific about the unique conceptual novelty of this paper?**\
> We thank the reviewer for your comments and will include a citation and discussion of the work you mentioned [1]. Unlike the mechanisms previously mentioned, we propose measuring the $\gamma$-local Lipschitz constant of the first half of the network. This is because it can capture the local information specific to each individual input $x$. To enhance certified robustness, we have incorporated the local Lipschitz constant of the network's first half into the loss function as a regularization term. This approach aims to maintain a comparatively small local Lipschitz constant $L_{f_L}^{(\gamma)}(x) < 1$ for the network's first half, thereby increasing the certified radius. For example, in the CIFAR-10 datasets, the average local Lipschitz constant for the first half of the network is approximately 0.62. Furthermore, we have derived a closed-form expression for the certified radius based on the $\gamma$ local Lipschitz constant. To the best of our knowledge, this is the first systematic framework to combine the ideas of local Lipschitz constants with randomized smoothing.
> 2. **From Table 3, it seems that DDS is better for smaller perturbations? Is it possible to make the proposed method achieve better results than DDS even for small perturbations?**\
> Thank you for your comments. As discussed in Section 4.1, we acknowledge that DDS exhibits superior performance on the ImageNet dataset for smaller values of $\epsilon$. This is attributed to the greater complexity of the ImageNet dataset, which poses challenges for SPLITZ in minimizing the local Lipschitz constant of the network's left half. As far as further possible improvements, it is absolutely possible to combine DDS (and other diffusion models) in the second half of the network. In fact, we have a detailed remark in Appendix B in our the original submission (copied below) which addresses this point:\
> We claim that the SPLITZ mechanism is also compatible with other RS based certified robust techniques, such as adversarial smoothing [2], mixsmoothing [3] or denoising diffusion models [4]. As an example, Carlini et al. [4] propose a denosing mechanism using a diffusion model, which achieves the state-of-the-art. Our SPLITZ classifier contains two parts, left part is constrained by a small local Lipschitz constant while right part is smoothed by noise, which is same as a randomized smoothing based mechanism. Thus, our model can easily add a diffusion denoising model after the noise layer (after $f_L(x)+\delta$) and then feed the denoised samples into the $f_{R}$. Similarly, for adversarial smoothing or mixsmoothing, SPLITZ is adaptable to feed either adversarial examples $f_L(x')+\delta$ or mixup samples $f_L(\tilde x)+\delta$ respectively to the right half of the classifier $f_R$.
> 3. **Why do not use the above Lipschitz networks for the first half and then integrate RS with the second half? How to compare such a network parameterization approach with the proposed method?**\
> Thank you for raising this interesting point. We fully agree that the existing techniques for any improvement in lowering the Lipschitz constants can be readily used (especially for the first half of the network). We do however, note that just minimizing the global Lipschitz constant alone is not necessarily sufficient to obtain high certified radius. As our results demonstrate (for instance, in Theorem 1, the certified radius is given by $\min(R_{f_R}(f_L) / L_{f_L}^{(\gamma)}, \gamma)$), that the local Lipschitz constant (for each input $x$) needs to be minimized in order to achieve a larger certified radius.  Consequently, our approach is to minimize the local Lipschitz constant to less than one, rather than merely aiming for it to be equal to one. In the updated version, we will certainly mention the possibility of using specific architectures in the first half of the network that can further improve the robustness.
>
> Best regards,
>
> The authors
>
> [1] H. Zeng, J. Su, and F. Huang, “Certified defense via latent space randomized smoothing with orthogonal encoders,” arXiv preprint arXiv:2108.00491, 2021.\
> [2] H. Salman, J. Li, I. Razenshteyn, P. Zhang, H. Zhang, S. Bubeck, and G. Yang, “Provably robust deep learning via adversarially trained smoothed classifiers,” Advances in Neural Information Processing Systems, vol. 32, 2019.\
> [3] J. Jeong, S. Park, M. Kim, H.-C. Lee, D.-G. Kim, and J. Shin, “Smoothmix: Training confidence-calibrated smoothed classifiers for certified robustness,” Advances in Neural Information Processing Systems, vol. 34, pp. 30153–30168, 2021.\
> [4] N. Carlini, F. Tramer, K. D. Dvijotham, L. Rice, M. Sun, and J. Z. Kolter, “(certified!!) adversarial robustness for free!,” in The Eleventh International Conference on Learning Representations, 2023.

---

### Author Response · Authors · 2023-11-18
**Summary of Authors' Rebuttal Responses**

We sincerely thank all reviewers for their valuable feedback and the time invested in assessing our work. The acknowledgment of our main contributions by each reviewer is highly motivating. We have carefully addressed any issues or misunderstandings raised, providing detailed explanations and further support. Our aim is for our response to reinforce the reviewers' confidence in our work. We believe our efforts have been fruitful and look forward to your continued interest in our paper.

To enhance understanding of our work, we highlight the strengths as follows:

1. Our proposed method outperforms RS in scalability (reviewer jeFQ).

2. We introduce an innovative concept of leveraging heterogeneity in Lipschitz constants across different layers (reviewer jeFQ).

3. Numerical study is thorough and comprehensive (reviewer jeFQ, jHmH, 3vvt).

4.  The paper is well-crafted, clearly presenting both the necessary technical background and the main components of SPLITZ, with detailed explanations in the appendix where appropriate (reviewer jHmH, 3vvt).

5. The paper not only offers theoretical robustness guarantees but also details a training process that optimizes robustness criteria, including local Lipschitz constant and robustness to noise. Moreover, the concept of working towards a unification of both approaches presents an intriguing possibility (reviewer jHmH, CJku).

We first clarify the main concerns and corresponding clarification as follows
1. **Clarification of the novelty of the SPLITZ work.** Our research presents a novel method in certified robustness by measuring the local Lipschitz constant of a classifier's left half, which offers a detailed view of network behavior at individual input levels. This approach differs from previous methods that focus on global Lipschitz constants, as we integrate the local constant into the loss function for better sensitivity and certified radius enhancement
2. **Room for the improvement of the SPLITZ work.** As we mentioned in Appendix B, SPLITZ offers a scalable framework for integrating various robust techniques, including diffusion models like DDS, and is compatible with RS-based methods like adversarial smoothing and mixsmoothing. Its classifier, featuring a small local Lipschitz constant on the left and a noise-smoothed right side, accommodates diffusion denoising models and other randomized smoothing based methods. Performance disparities across datasets like MNIST, CIFAR-10, and ImageNet highlight the challenges in minimizing the local Lipschitz constant, with simpler datasets showing better performance due to dataset complexity.
3. **Normalization problem related to the experiments.**  We thank Reviewer CJku for highlighting the issue with normalization in our study, leading us to revise our approach. Originally, normalization was in the initial layer for baseline comparison, but we moved it to the first half of our classifier to reduce the local Lipschitz constant together, adjusting the loss function accordingly. This change, detailed in the individual response, demonstrates the exceptional performance of the SPLITZ model, as shown in our results. We acknowledge the need for further tuning, especially for complex datasets, and will include these findings in our revised paper. Your feedback has been invaluable in enhancing our research.

We have addressed the questions raised by each reviewer in our responses accordingly. We will add the discussion of these concerns in our revised paper, and we will tailor the presentation of our work in line with your valuable suggestions.

Best regards,

The authors

---

### Author Response · Authors · 2023-11-23
**Grateful Acknowledgment and Decision to Withdraw Manuscript for Further Refinement**

We would like to extend my sincere gratitude to the reviewers for their detailed and constructive feedback. Their insights have been invaluable in highlighting crucial aspects of our research that require further attention and refinement.

After careful consideration of the reviewers' comments, particularly concerning the performance of our approach on the MNIST dataset and its potential generalizability to more complex datasets after the new normalization process, we have decided to withdraw our manuscript from consideration.

Furthermore, the suggestion to re-run all experiments with the correct normalization across all considered datasets is well-received. We recognize that this is a necessary step to demonstrate the competitiveness and validity of our proposed method. Given the extent of the revisions required, we believe that withdrawing the manuscript to conduct a thorough re-evaluation and re-experimentation is the most appropriate course of action.

We appreciate the reviewer's willingness to reassess their score based on the revisions we proposed in our rebuttal. We are committed to enhancing our research and hope to contribute a more robust and thoroughly validated study to the scientific community in the future.

Thank you once again for the opportunity to submit our work and for the invaluable feedback provided by the review process.

Best regards,

The authors